# RESILIENCE TO MULTIPLE ATTACKS VIA ADVERSARIALLY TRAINED MIMO ENSEMBLES

## ABSTRACT

While ensemble methods have been widely used for robustness against random perturbations (*i.e.,* the average case), ensemble approaches for robustness against adversarial perturbations (*i.e.,* the worst case) have remained elusive despite multiple prior attempts. We show that ensemble methods can improve adversarial robustness to multiple attacks if the ensemble is *adversarially diverse*, which is defined by two properties: 1) the sub-models are adversarially robust themselves against multiple attacks, and yet 2) sub-models are diverse measuring by adversarial transferability. While at first glance, creating such an ensemble would seem computationally expensive, we demonstrate that an adversarially diverse ensemble can be trained with minimal computational overhead via a Multiple-Input Multiple-Output (MIMO) model. Specifically, we propose to train a MIMO model with adversarial training (*MAT*), where each sub-model can be trained on a different attack type. When computing gradients for generating adversarial examples during training, we use the gradient with respect to the ensemble objective. The benefits of this scheme are twofold: 1) it only requires 1 backward pass and 2) the cross-gradient information between the models promotes robustness against transferable attacks. We empirically demonstrate that *MAT* produces an ensemble of models that is adversarially diverse and significantly improves performance over single models or vanilla ensembles while being comparable to previous state-of-the-art methods. On MNIST, we obtain $99.5\%$ clean accuracy and $(82.3\%, 57.1\%, 71.6\%)$ against $(\ell_\infty, \ell_2, \ell_1)$ attacks, and on CIFAR10, we achieve $79.7\%$ clean accuracy and $(47.9\%, 61.8\%, 47.6\%)$ against $(\ell_\infty, \ell_2, \ell_1)$ attacks, which are comparable to previous state-of-the-art methods.

## 1 INTRODUCTION

The arms race between attacks and defenses against DNNs has created a diversity of attack types. These include black-box vs white-box attacks and gradient-based vs gradient-free attacks. Ideally, adversarially robust models would be robust against all types of attack. Many defenses have been developed to protect DNNs from adversarial attacks. For instance, gradient masking [21], input denoising [33, 15], and hidden layer randomization [16], are three techniques that in their time were heralded as strong defense techniques. However, [1, 3, 28] show that all of the above methods fail against adaptive adversaries, *i.e.,* perturbations crafted with full knowledge of the defense measure. Among all attempts to make the deep model resilient, Adversarial Training (AT) [17] has stood the test of time and is regarded as the gold standard for its resilience to white-box adversaries [18]. AT trains the deep models with adversarial examples instead of with benign training examples. Due to its empirical robustness, this method remains state-of-the-art against a wide range of attacks. However, AT suffers from two fundamental drawbacks. First, it is computationally expensive, specifically the method to generate effective AEs. Second, the choice of attack type used during training significantly affects the type of robustness that is observed. For example, previous work [27, 22] shows that the model adversarial trained using the $\ell_\infty$ attack is robust against $\ell_1$ and $\ell_2$ PGD attacks but vulnerable

against gradient-free attacks. This phenomenon reveals that the robustness of adversarial training implicitly masks out the gradient information for $\ell_1$ and $\ell_2$ attacks. Previous works [27, 18] try to revise adversarial training by training the model using the worst case of PGD adversarial examples among different attack metrics and achieve better results. MSD [18] revises the Tramèr and Boneh's method [27] by calculating $\ell_1, \ell_2, \ell_\infty$ attacks for *each step* of PGD iteration and updates with the one that achieves the largest loss. This method achieved better performance against $\ell_1, \ell_2, \ell_\infty$ attacks. However, though MSD is faster than [27], it still requires multiple forward passes for each step of the PGD attack, which makes it more expensive to train comparing to the standard AT.

As ensemble methods have proven to be effective in robustness to random perturbations (*i.e.,*, the average case), one natural idea to improve adversarial robustness is to exploit ensemble methods (or model averaging). Ensemble methods combine the predictions of sub-models based on some vote functions (like averaging or majority vote). Previous attempts of using ensemble methods for adversarial robustness have been shown to be vulnerable under strong adversaries [35, 19, 23, 29, 10]. They mainly failed for two reasons [34]. First, if the ensemble is diverse but the models are weak (*i.e.,* not adversarially robust), an adaptive attack only needs to find a *single* AE that lies in the intersection of the AE space of the sub-models (*i.e.,* a single worst case), which is easy [10]. Intuitively, this finds a single perturbation that is a "common mistake" across sub-models. Second, adversarial examples tends to transfer between similar models, *i.e.,* the adversarial perturbation generated from one sub-model may easily mislead another sub-model especially when they share the similar training set or architecture [20]; and the transferability of attacks remains true even if the sub-models are adversarially robust themselves (see vanilla ensemble models in 4.2 of the experiments). Thus, neither diversity of the models nor adversarial robustness of each of the sub-models by themselves can improve adversarial robustness.

To overcome these two prior failures of ensemble methods, we propose to encourage *adversarial diversity* which is defined to have two key features simultaneously: 1) adversarial robustness of each sub-model and 2) robustness against transfer attacks between sub-models. Additionally, we would like to do this with minimal computational overhead because AT is already very computationally expensive. Our approach is inspired by a recent work called multi-input-multi-output (MIMO) training [8]. MIMO can train $M$ sub-models simultaneously to form a diverse ensemble, and crucially the training time of the ensemble is almost the same as that of a single model. In this work, we show that by combining MIMO configuration with AT, it is possible to attain robustness against multiple attack types with an adversarially diverse ensemble, i.e., we can achieve adversarial diversity. The results in § 4.1 shows that by sharing model parameters, the sub-models act as an implicit way of bringing adversarial diversity while also remaining robust. Then in § 4.2, we present that our *MAT* adversarially trained with each sub-model bestowing resilience to a particular attack metric leads to robustness against multiple types of perturbations. We achieve robustness accuracy $(82.3\%, 57.1\%, 71.6\%)$ against $(\ell_\infty, \ell_2, \ell_1)$ attack with epsilon $(10, 2, 0.3)$ on MNIST. On CIFAR10 dataset, our model achieves robustness accuracy $(47.9\%, 61.8\%, 47.6\%)$ against $(\ell_1, \ell_2, \ell_\infty)$ attack with radius $(12, 0.5, 0.03)$. Finally, in § F, we provide an alternative application that one can utilize our MIMO+AT (*MAT*) by training the *MAT* with all sub-models trained with the same attack to strengthen previous AT methods to achieve better adversarial robustness without loss of clean accuracy. We provide three examples in this section as $\ell_\infty$ AT, $\ell_2$ AT, and MSD. We summarize our contributions as follows:

**1.** We shed light on why prior ensemble methods may fail and hypothesize that ensemble methods can improve adversarial robustness if they are *adversarially diverse*, which is defined by two properties: 1) the sub-models are adversarially robust themselves and yet 2) adversarial attacks do not transfer easily between sub-models. **2.** We propose to use MIMO AT (*MAT*) to train such an ensemble in a computationally efficient way and discuss how the cross-gradient information between the sub-models during training can increase the robustness to transfer attacks between models. **3.** We empirically demonstrate that *MAT* is adversarially diverse and improves significantly over AT models and vanilla ensembles while being comparable or better than prior state-of-the-art methods for multi-attack robustness.

## 2 PRELIMINARY

In this section, we review some key concepts related to adversarial attacks, adversarial training (AT), and MIMO structure, all the notations defined here will be used throughout the paper.

**PGD Adversarial Attacks.** PGD attacks seek a point $x'$ in a neighborhood of the benign example $x$ with perturbation $\delta$, which maximizes the loss $\mathcal{L}$. $x'$ is generated as: $x' = x + \arg\max_{\delta \in \mathcal{B}_{\|\cdot\|_p}(\epsilon)} \mathcal{L}(f_\theta(x+\delta), y^*)$, where $\epsilon$ denotes the attack radius, $f_\theta$ is the DNN with parameter $\theta$, $y^*$ is the true label. Denote the perturbation $\delta = x' - x$. A general PGD attack adopts gradient-type algorithm to update $\delta$ as

$$\delta^{(t+1)} = \Pi_{\mathcal{B}_{\|\cdot\|_p}(\epsilon)}\left(\delta^{(t)} + \alpha \cdot \mathcal{V}_p\left(\nabla_x \mathcal{L}(f_\theta(x), y^*)\right)\right), \quad \text{with} \quad \mathcal{V}_p(\delta) = \arg\max_{v \in \mathcal{B}_{\|\cdot\|_p}(1)} v^\top \delta. \quad (1)$$

where $p$ is the attack metric (usually $\ell_p$ norm), $\alpha$ is the step size, $\Pi$ is the operator projecting the gradient onto the $\ell_p$ ball with radius $\epsilon$ (denoted by $\mathcal{B}_{\|\cdot\|_p}(\epsilon)$), $\mathcal{V}_p$ is the operator mapping the gradient to the steepest descent direction on the $\ell_p$ ball with radius 1. Note that $\mathcal{V}_p$ is a fundamentally different operator than $\Pi$, and in the case of the $p = \infty$, $\mathcal{V}_\infty$ reduces to the sign of the gradient (as in the FGSM attack). Intuitively, PGD adversarial attack uses iterate with a fixed step size $\alpha$ along the steepest direction $\mathcal{V}_p\left(\nabla_x \mathcal{L}(f_\theta(x), y^*)\right)$. FGSM [7], BIM [13] and [17] can all be treated as variants of the PGD Attack.

**Adversarial Training.** AT is regarded as one of the most practical yet effective methods for defending against adversarial attacks. This has stood the "test of time" through multiple iterations of proposed defenses and attacks that discover vulnerabilities in them. AT seeks to build a robust model against adversarial attacks by producing adversarial examples and injecting them into training data by solving a saddle point optimization problem

$$\theta^* = \arg\min_\theta \sum_i \max_{\delta_i \in \mathcal{B}_{\|\cdot\|_p}(\epsilon)} \mathcal{L}\left(f_\theta(x_i + \delta_i), y_i^*\right). \quad (2)$$

Here $i$ denote the $i$-th batch number of training set $(x_i, y_i^*)$. The inner maximization seeks to find the adversarial directions $\delta_i$ given the current classifier $f_\theta$, and the outer minimization tries to find the best $\theta$ to minimize the summation of all batches' loss. PGD AT is widely accepted as a standard approximation of inner maximization methods since it is a universal first-order attack algorithm [17]. Though it roughly needs $K + 1$ times more computation compared to standard training, PGD-AT remains the de facto standard for AT since it walks the balance between strength of attack and computational cost of generating adversarial examples. The drawback of AT is that solving the inner maximization requires a pre-defined attack metric $p$, which only contributes to robustness under $\ell_p$ attacks. [27]. In this paper, we show how the adversarial robustness can be achieved through AT but crucially, one can generalize the defense to multiple attack types and achieve this at about the exact cost as AT.

**MIMO Ensemble.** Ensemble methods have been widely used to improve a model's calibration and robustness over out-of-distribution samples by averaging over multiple neural network predictions [8, 31, 32]. However, ensemble over $M$ deep models (sub-models) is usually $M$ times more expensive from a computational and a memory standpoint. Furthermore, to improve uncertainty estimation, sub-models often sacrifice individual performance to achieve diversified predictions.

To address the computational drawbacks of vanilla ensembles, [8] proposed an ensemble structure called MIMO Ensemble. It showed a surprising result: the benefits of using multiple predictions can be achieved 'for free' under a single model's forward pass. In particular, they showed that, using a multi-input multi-output (MIMO) configuration, one can utilize a single model's capacity to train multiple sub-models that independently learn the task at hand. Since all the layers are shared except for the input layer and the output layer, this architecture uses only about 1% more parameters. Training a MIMO model has almost

the same time complexity as training a standard model while keeping the sub-models diverse without any special regularization and parameter tuning. During training, MIMO uses gradient method to solve following optimization problem for $M$ sub-models.

$$\underset{\theta}{\text{minimize}} \sum_i \sum_{m=1}^{M} \mathcal{L}(\underbrace{f_\theta(x_{i,1}, ..., x_{i,M})_m}_{y_{i,m}}, y_{i,m}^*), \qquad (3)$$

where $(x_{i,1}, ...x_{i,M})$ are i.i.d sampled from the training set, with $x_{i,m}$ denoting the input of the $i$-th batch for the $m$-th subnetwork, $y_{i,m}, y_{i,m}^*$ represent the outputs from the model and the ground truth respectively, and $\mathcal{L}$ is the cross entropy loss. Notice that the overall objective is the summation of each sub-model's loss. The separation structure promotes the sub-models' prediction to be less correlated because the output $y_j$ is independent to $x_i$ when $i \neq j$. During testing, the same inputs are copied and fed to all the $M$ sub-models, then MIMO decides the prediction by averaging $y_{i,m}$.

## 3 OUR APPROACH: *MAT*

Given the weaknesses of prior ensemble methods, we seek an algorithm that produces an *adversarially diverse* ensemble, where the sub-models are robust to the attacks transferred from other sub-models so that the ensemble can have better robustness comparing to single sub-model. To achieve this, we propose to combine MIMO and AT, an approach we call *MAT*.

There are two difficulties to overcome in adversarially training an ensemble model: 1) For the outer minimization, how should one choose the loss function to update the sub-model parameters $\theta$? To be specific, choosing a single sub-model loss or the ensemble loss? 2) For the inner maximization, how should the adversarial examples $x'_1, \ldots, x'_M$ be created during training? To solve (1), [8] demonstrates that taking the gradient with respect to the ensemble loss contributes to higher clean accuracy *and* is more computationally efficient. Therefore, we choose to update parameter $\theta$ using SGD, where the gradient is taken with respect to the ensemble loss instead of the loss of each sub-model. To tackle (2), one natural choice is taking gradient with respect to *each sub-model's loss* as in standard AT on each sub-model. However, we show in Sec. 3.2 that taking gradients with respect to the *ensemble* loss produces sub-models that re more robust to transfer attacks and, furthermore, it is more computationally efficient. Therefore, in the inner maximization, we generate adversarial examples by algorithms that compute gradients with respect to the *ensemble* loss instead of each sub-model's loss. We explore these two choices in more detail in the next two sections.

### 3.1 NAÏVE MIMO ADVERSARIAL TRAINING

The foundation of *MAT* is MIMO adversarial training method. The goal is to train an ensemble that is adversarially diverse. To encourage diversity, we use different attack types for each sub-model. We adopt the common choice in the literature as three PGD attacks $\ell_1, \ell_2$, and $\ell_\infty$ to instantiate our network. We aim to solve the following problem

$$\min_\theta \sum_i \sum_{m=1}^{M} \max_{\delta_{i,m} \in \mathcal{B}_{\|\cdot\|_{p_m}}(\epsilon_m)} \mathcal{L}(\underbrace{f_\theta(x_{i,1}+\delta_{i,1}, ..., x_{i,M}+\delta_{i,M})_m}_{y'_{i,m}}, y_{i,m}^*), \qquad (4)$$

where for each batch $i$, and each sub-model $m$, the inner maximization seeks the worst-case adversarial perturbation $\delta_{i,m} \in \mathcal{B}_{\|\cdot\|_{p_m}}(\epsilon_m)$ under the current classifier $f_\theta$. Notice that $\delta_{i,m}$ might be different for a different sub-model $m$. As a naïve approach, we use $K$-step PGD to update $\delta_{i,m}^k$ as the following

$$\delta_{i,m}^{(k+1)} = \Pi_{\mathcal{B}_{\|\cdot\|_{p_m}}(\epsilon_m)} \left( \delta_{i,m}^{(k)} + \alpha_m \cdot \mathcal{V}_{p_m} \left( \nabla_{x_{i,m}} \mathcal{L}(y'_{i,m}, y_{i,m}^*) \right) \right). \qquad (5)$$

Notice the difference between equation 5 and equation 1: for different $m$, equation 5 allows for different projections $\mathcal{B}_{\|\cdot\|_{p_m}}$, stepsize $\alpha_m$ as well as steep descent direction $\mathcal{V}_{p_m}$, which results in potential robustness over diverse attacks; however, equation 1 can only be seen as special case of equation 5 with $m = 1$, therefore it lacks multiple robustness after the training.

### 3.2 MIMO ADVERSARIAL TRAINING (MAT)

While the naïve MIMO adversarial training may produce models that are adversarially robust by themselves, it does not prevent adversarial examples that would transfer across multiple sub-models and, additionally, it is computationally expensive since it requires $M$ backward passes for each $\delta$ computation. Therefore, to further improve the adversarial diversity of *MAT*, and also reduce the computational burden, we propose to replace equation 5 by the following

$$\delta_{i,m}^{(k+1)} = \Pi_{\mathcal{B}_{\|\cdot\|_{p_m}}(\epsilon_m)} \left( \delta_{i,m}^{(k)} + \alpha_m \cdot \mathcal{V}_{p_m} \left( \nabla_{x_{i,m}} \sum_{j=1}^{M} \mathcal{L}(y'_{i,j}, y^*_{i,j}) \right) \right), \tag{6}$$

where for each tuple $(i, m)$, we used the approximate gradient $\nabla_{x_{i,m}} \sum_{j=1}^{M} \mathcal{L}(y'_{i,j}, y^*_{i,j})$ instead of the true gradient $\nabla_{x_{i,m}} \mathcal{L}(y'_{i,m}, y^*_{i,m})$, where the difference between the two is defined as the cross gradient

$$\text{cross gradient } (i, m) \triangleq \nabla_{x_{i,m}} \sum_{j \neq m} \mathcal{L}(y'_{i,j}, y^*_{i,j}) = \nabla_{x_{i,m}} \sum_{j=1}^{M} \mathcal{L}(y'_{i,j}, y^*_{i,j}) - \nabla_{x_{i,m}} \mathcal{L}(y'_{i,m}, y^*_{i,m}). \tag{7}$$

We include the pseudo-code of MAT in appendix Sec. A.1, where the cross loss related to equation 7 allows the exchange of information between the networks during the attack stage which may help in reducing the transferability of Adversarial examples. It also plays an essential role in reducing the computation burden as well as improving the adversarial diversity.

To obtain all the perturbations $\delta_{i,m}, m = 1, \ldots, M$ using the true gradient update equation 5, we need $M$ times backward propagation (time complexity analysis is in the appendix B). However, we only need 1 backward propagation when we adopt the approximate gradient step equation 6. Therefore equation 6 reduces the computational burden dramatically.

Next, we explore two properties of the cross gradient which contribute to adversarial diversity: 1) **Reduced adversarial examples transferability between sub-models:** In Section 2, we showed that minimization weakens the correlation between the sub-models due to MIMO structure. However, for problem equation 5, there's still a possibility that the adversarial perturbation $\delta_{i,m}$ could flip the output $y_j$ of the $j$-th sub-model. Due to minimax structure of the problem equation 5, the flipping phenomena can be effectively reduced by introducing the cross loss during maximization training process, where the cross gradient $(i, m)$ describes the change of cross loss $\sum_{j \neq m} \mathcal{L}$ w.r.t the change of $x_{i,m}$ locally. Therefore, introducing cross loss can improve the robustness of MAT against different attacks. 2) **Cross-gradients become small but non-zero during training:** This can be validated by Fig. 1, where the cross gradients become very small during the training but the final cross gradient does not become exactly 0, showing that there is still weak correlation between sub-models.

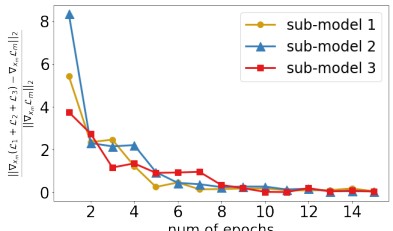

Figure 1: Training: validating the cross gradients norm approaching to zero: the number of epochs versus the ratio between $\ell_2$-norm of the cross gradient term and the $\ell_2$-norm of the true gradient.

### 4 EVALUATION

In § 4.1, we demonstrate that the sub-models trained by our *MAT* are more robust to adversarial examples generated from other sub-models (reduced adversarial transferability). Furthermore, we show that each sub-model is robust against multiple adversarial examples. In § 4.2, we present experimental results on *MAT* versus state-of-the-art multi-robustness defenses for a variety of attacks.

## 4.1 DIVERSITY AND ROBUSTNESS

**Diversity of adversarial transferability among sub-models** A group of classifiers can benefit from the ensemble if and only if each member is diverse enough. There are several different measures trying to quantify the property [12, 35]. In this section, we observe how the ensemble of sub-models perform against diverse adversarial attacks. To explore the diversity among sub-models of vanilla-ensemble (resp. *MAT*), the sub-models 1, 2, and 3 have been trained against respectively $\ell_1$, $\ell_2$, and $\ell_\infty$ PGD attacks; then, we evaluate vanilla-ensemble (resp. *MAT*) against $\ell_\infty$-adversarial attacks and report their clean accuracy (the results corresponds to $\ell_1$ and $\ell_2$ are similar and summarized in appendix G We study two kinds of $\ell_\infty$-adversarial examples. The first kind is $\ell_\infty$-adversarial attack generated from sub-model 1, from which we can study the transferability of $\ell_\infty$-adversarial attack generated from sub-model 1 by counting the successful attacks to sub-model 2, sub-model 3, and vanilla-ensemble (resp. *MAT*). In particular, we denote

$$N_{ii} \triangleq \#\{\text{successful attacks to sub-model } i \text{ out of 1000 } \ell_\infty\text{-attacks generated from sub-model } i\},$$

$$N_{ij} \triangleq \#\{\text{successful attacks to sub-model } j \text{ out of 1000 } \ell_\infty\text{-attacks generated from sub-model } i\}.$$

Then, we define the transferability of adversarial examples from $i$ to $j$ as $r_{ij} \triangleq N_{ij}/N_{ii}$. Furthermore, we use $\ell_\infty$-adversarial examples generated from vanilla-ensemble (resp. *MAT*) to attack each sub-model and their vanilla-ensemble (resp. *MAT*) to evaluate the overall performance of vanilla-ensemble (resp. *MAT*). The results are summarized in second row of Table. 1 (resp. Table. 2) for vanilla-ensemble (resp. *MAT*).

Table 1: Vanilla-Ensemble diversity testing: successful $\ell_\infty$-adversarial attack. Each column shows the misclassification samples out of 1000. The first (resp. second) row corresponds to $\ell_\infty$ adversarial attacks generated from model 2 (resp. vanilla-ensemble). The last row relates to clean accuracy with no attack.

|  | Sub-model 1 | Sub-model 2 | Sub-model 3 | Vanilla-Ensemble |
|---|---|---|---|---|
| Sub-model 1 | 39 | 78(200%) | 120(307.7%) | 60(153.8%) |
| Vanilla-Ensemble | 45 | 994 | 998 | 960 |
| No Attack | 14 | 9 | 12 | 12 |

Table 2: *MAT* diversity testing: successful $\ell_\infty$ adversarial attack. Each column shows the misclassification samples out of 1000. The first (resp. second) row corresponds to $\ell_\infty$ adversarial attacks generated from sub-model 2 (resp. *MAT*). The last row relates to clean accuracy with no attack.

|  | Sub-model 1 | Sub-model 2 | Sub-model 3 | MAT |
|---|---|---|---|---|
| Sub-model 1 | 156 | 37(23.7%) | 52(33.3%) | 64(41.0%) |
| *MAT* | 114 | 548 | 106 | 128 |
| No Attack | 9 | 7 | 7 | 5 |

Comparing Table. 1 and Table. 2, we observe the following phenomena. First, the transferability ratios of *MAT* are $r_{12} = 23.7\%$, $r_{13} = 33.3\%$, which are much lower than $r_{12} = 200\%$, $r_{13} = 153.8\%$ of vanilla ensemble. This indicates the sub-models of *MAT* are more robust to $\ell_\infty$-attacks, and $\ell_\infty$-attacks are easier to transfer between the sub-models of vanilla ensemble. Therefore, we conclude that the sub-models of *MAT* are diverse enough to reduce the effective $\ell_\infty$ adversarial transferability. Second, the performance of vanilla-ensemble is close to the worst sub-model performance. This is indicated by noticing the number of successful $\ell_\infty$-attacks for vanilla-ensemble is 960, which is close to $\max\{45, 994, 998\} = 998$. This means that vanilla-ensemble does not provide extra ro-

Table 3: Testing: Robustness accuracy of *MAT* and its sub model: each row corresponds to different attacks; and each column depicts the robustness accuracy of *MAT* and its sub model.

|  | *MAT* | sub 1 | sub 2 | sub 3 |
|---|---|---|---|---|
| $\ell_\infty$ PGD | 89.2% | 85.0% | 33.1% | 87.1% |
| $\ell_2$ PGD | 66.6% | 63.7% | 69.0% | 61.0% |
| $\ell_1$ PGD | 71.9% | 72.0% | 71.4% | 67.8% |

bustness compared to its sub-models. However, the successful $\ell_\infty$-attack against *MAT* is only 128, which

only takes up $23.4\%$ of the worst sub-model performance $\max\{114, 548, 106\} = 548$. This suggests that the *MAT* clearly improves the robustness accuracy compared to that of sub-models.

**Robustness of sub-models** We empirically show that the sub-models of *MAT* are robust against $\ell_\infty, \ell_2, \ell_1$ attacks. During testing, the same inputs are copied and fed to all the $M$ sub-models, then *MAT* decides the prediction by averaging $y_{i,m}$. The results are summarized in Table 3. We observe that using *MAT*, each sub-model is relatively robust against all adversarial examples. The only weak sub-model is sub-model 2, which shows vulnerability against $\ell_\infty$ PGD attack ($33.1\%$), however, *MAT* still performs well against $\ell_\infty$ attacks ($89.2\%$) because it averages the output of the three sub-models.

## 4.2 MAIN RESULT

In this section, we present two main results of *MAT* on MNIST and CIFAR-10 against various types of attacks and compare it with different baselines. *MAT* generally boosts the previous methods without extra cost.

**Main result on MNIST** Our main results on MNIST are summarized in Figure 2 and Table 4. Figure 2 plots varying number of attack steps versus robust accuracy, and the varying radius using $\ell_\infty$ (resp. $\ell_2$ and $\ell_1$) versus robust accuracy. We can capture the following phenomena: (1) both of MAT and MAT+MSD exhibit generalization and robustness for a varying number of attack steps; (2) MAT outperforms MSD and MAT+MSD depending against $\ell_\infty$- PGD attack; and when it comes to $\ell_2$-PGD attack and $\ell_1$-PGD attack, the performance of MAT+MSD surpass the other two methods on MNIST for a varying number of attacks; (3) MAT outperform MSD on $\ell_\infty$- PGD attack; 2) on $\ell_2$- (resp. $\ell_1$-) PGD attack, MAT+MSD surpass the other two method. Table. 4 shows under the attack radius choice 0.3 (resp. 2 and 10) for $\ell_\infty$ (resp. $\ell_2$ and $\ell_1$),

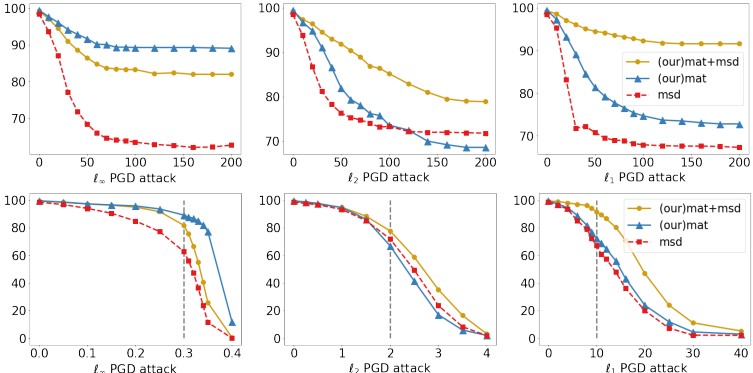

Figure 2: Robust accuracy curves of *MAT* +MSD, *MAT*, and MSD on MNIST under PGD $\ell_\infty$ (resp. $\ell_2$, and $\ell_1$) attack. **First row:** Robust accuracy against the number of steps of attack with radius 0.3 (resp. 2, and 10). **Second row:** Robust accuracy against varying attack radius using $\ell_\infty$- (resp. $\ell_2$ and $\ell_1$) PGD attack; the vertical dashed lines denotes the common radius choice in prior work evaluations as 0.3 (resp. 2 and 10) for $\ell_\infty$ (resp. $\ell_2$ and $\ell_1$).

the robustness accuracy comparison under different attack metrics between *MAT* and AT, vanilla ensemble MSD, standard training is listed for clean accuracy comparison[1]. All $\ell_\infty$ (resp. All $\ell_2$ and All $\ell_1$) attack is to defend against $\ell_\infty$-based attacks, *i.e.,* the union of PGD $\ell_\infty$ attack, HSJ attack, Auto Attack (resp. PGD $\ell_2$ attack, C&W attack, Boundary attack for $\ell_2$- based attacks; and PGD $\ell_1$ attack, Salt & Pepper attack for $\ell_1$-

---
[1]All the hyperparameter including the step size of the PGD attack, attacks radius $\epsilon$, and termination max iteration are chosen as the best tuning for MSD [18] in order to present results on a same base line (which may not be the best for MAT). Please check Sec. C. for detailed information about hyperparameters choice.

based attacks). This reveals that *MAT* could improve the overall model's robustness. The vanilla ensemble shows its vulnerability to all three kinds of adversarial attacks though it combines a three times larger model and requires three times longer training. *MAT* achieves $82.3\%$ against the union of $\ell_\infty$ attacks on the MNIST dataset, which surpasses the previous state-of-the-art by $26.0\%$ percent. For defending against the union of adversarial attacks, our method attains $51.7\%$ robustness accuracy.

Table 4: Testing on MNIST: each row corresponds to different attacks; each column compares the robustness accuracy of *MAT* with different baseline models.

|  | standard training | $\ell_\infty$ | $\ell_2$ | $\ell_1$ | Vanilla Ensemble | MSD | MAT (ours) |
|---|---|---|---|---|---|---|---|
| clean accuracy | 99.5% | 99.3% | 98.6% | 98.9% | 99.0% | 98.5% | **99.5%** |
| PGD $\ell_\infty$ attack | 0.0% | 96.0% | 0.8% | 0.0% | 4.6% | 62.2% | 89.2% |
| HSJ attack | 12.3% | 98.0% | 44.2% | 12.1% | 35.7% | 87.4% | 91.2% |
| Auto Attack | 0.1% | 94.9% | 0.5% | 0.2% | 0.1% | 57.0% | 82.3% |
| All $\ell_\infty$ attacks | 0.0% | 94.9% | 0.4% | 0% | 0.0% | 56.3% | 82.3% |
| PGD $\ell_2$ attack | 0.0% | 78.1% | 76.8% | 11.4% | 57.2% | 72.0% | 66.6% |
| C&W attack | 0.0% | 39.7% | 76.8% | 12.6% | 60.1% | 69.8% | 59.4% |
| Boundary attack | 0.7% | 17.7% | 78.2% | 21.2% | 46.9% | 70.6% | 77.8% |
| All $\ell_2$ attacks | 0.0% | 8.7% | 74.4% | 9.8% | 44.8% | 67.3% | 57.1% |
| PGD $\ell_1$ attack | 0.0% | 53.3% | 79.0% | 19.9% | 63.2% | 63.7% | 71.9% |
| Salt & Pepper attack | 74.3% | 64.9% | 96.2% | 72.5% | 68.4% | 84.5% | 96.0% |
| All $\ell_1$ attacks | 0.0% | 53.3% | 78.2% | 19.9% | 60.1% | 62.7% | 71.6% |
| All attacks | 0.0% | 6.5% | 0.8% | 0.0% | 1.5% | 54.7% | 51.7% |

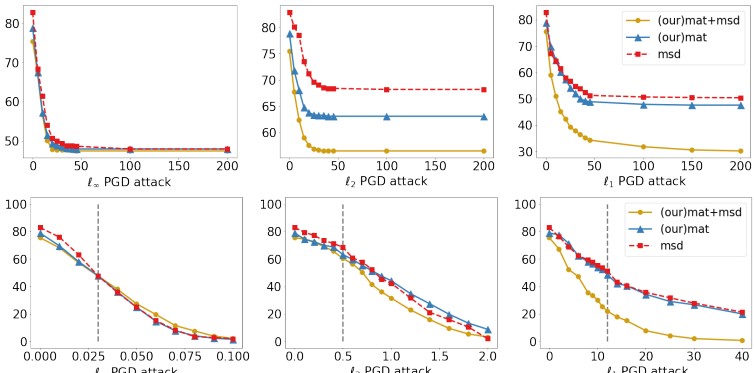

Figure 3: Robust accuracy curves of *MAT* +MSD, *MAT*, and MSD on CIFAR-10 under PGD $\ell_\infty$ (resp. $\ell_2$, and $\ell_1$) attack. **First row:** Robust accuracy versus the number of attack steps $0.03$ (resp. $0.5$, and $12$). **Second row:** Robust accuracy versus the varying radius using $\ell_\infty$ (resp. $\ell_2$ and $\ell_1$) PGD attack; the vertical dashed lines denote the common radius choice in testing as $0.03$ (resp. $0.5$ and $12$) for $\ell_\infty$ (resp. $\ell_2$ and $\ell_1$).

**Main result on CIFAR-10.** Our main results on CIFAR-10 are summarized in Figure 3 and Table 5. We observe the following: (1) the performance of MAT is comparable w.r.t MSD on $\ell_\infty$ and $\ell_1$- PGD attack with varying steps and varying radius; (2) MSD exhibits better testing accuracy than the other two methods on CIFAR-10 for a varying number of $\ell_2$-PGD attacks with radius $r = 0.5$, however, MAT outperforms MSD when the attack radius increases (greater than the common choice $0.5$).

Table 5 shows that *MAT* achieve strong robustness against multiple kinds of attack. Crucially, the vanilla ensemble fails against the $\ell_\infty$ attacks while *MAT* is competitive against the state-of-the-art MSD, while being computationally more efficient ($K + 1$ forward passes for *MAT* versus $(1 + M)K + 1$ for MSD, with the same number of backward passes $K + 1$). We include the time complexity analysis in Table 6 and training time on each model in Table 7 please see appendix B and appendix E for details.

Table 5: Testing on CIFAR-10: each row corresponds to different attacks; each column compares the robustness accuracy of *MAT* with different baseline models.

| | standard training | $\ell_\infty$ | $\ell_2$ | $\ell_1$ | Vanilla Ensemble | MSD | MAT (ours) |
|---|---|---|---|---|---|---|---|
| clean accuracy | 95.3% | 84.9% | 89.3% | 93.6% | 84.9% | 81.1% | 78.9% |
| PGD $\ell_\infty$ attack | 0.0% | 48.8% | 0.0% | 0.0% | 0.0% | 48.0% | 47.9% |
| HSJ attack | 7.5% | 79.3% | 69.0% | 12.9% | 67.3% | 72.4% | 74.7% |
| Auto Attack | 0.1% | 78.3% | 0.0% | 0.0% | 0.0% | 57.0% | 56.4% |
| All $\ell_\infty$ attacks | 0.0% | 46.7% | 0.0% | 0.0% | 0.0% | 48.0% | 47.9% |
| PGD $\ell_2$ attack | 4.3% | 61.1% | 79.5% | 0.0% | 61.2% | 68.2% | 64.5% |
| C&W attack | 3.4% | 60.6% | 79.3% | 0.0% | 57.5% | 64.7% | 63.3% |
| Boundary attack | 25.2% | 19.4% | 82.1% | 1.6% | 23.2% | 69.2% | 64.2% |
| All $\ell_2$ attacks | 1.9% | 18.2% | 73.3% | 0.0% | 19.1% | 64.3% | 61.8% |
| PGD $\ell_1$ attack | 13.1% | 18.2% | 34.0% | 0.0% | 14.2% | 53.4% | 47.6% |
| Salt & Pepper attack | 53.5% | 65.7% | 75.4% | 61.3% | 65.6% | 73.9% | 75.0% |
| All $\ell_1$ attacks | 12.8% | 18.0% | 33.3% | 0.0% | 14.2% | 53.4% | 47.6% |
| All attacks | 0.0% | 16.2% | 0.0% | 0.0% | 0.0% | 47.0% | 46.4% |

## 5 RELATED WORK

**Adversarial Defense.** Many defenses have been proposed, with most of them being broken by adaptive attacks [1, 3, 28]. Adversarial Training [17], though expensive in computation, has emerged as the best practical way to create models resilient to white-box attacks. A slew of literature has developed variants of AT differing in the way samples are created [6], the way the models are iterated upon [24, 30], and the way different attacks are combined [27, 18]. Our ensemble method is orthogonal to these approaches and therefore can be combined with these, with varying levels of additional design and engineering effort, such as to design the right combining function across the sub-models.

**Adversarial Defense against multiple types of perturbation.** [18] tries to build a robust model against multiple perturbations by introducing multiple steepest direction PGD attacks during training; their method ensembles three different attacks into one, called *multi steepest direction*. Our method instead seeks an ensemble over *models*. [26] proposed a provable defense against all $\ell_p$ norm with robust radius $3 - 5$ times smaller than state-of-the-art heuristic defense methods [18]. This technique has the benefit that it uses transfer learning and therefore does not need separate training for each kind of attack. This can be used seamlessly with *MAT* by starting each sub-network after transfer learning from the another sub-network. [14] trains DNNs against imperceptible adversarial examples, which are approximated using DNNs. While it shows robustness against a wide range of attacks, its performance against specific attacks is limited. Our approach *MAT* outperforms prior solutions in this space, either as we have proven empirically for the latest approach [18] or through the author-published numbers on the same dataset.

## 6 CONCLUSION AND DISCUSSION

In this paper, we present an adversarially diverse ensemble method called *MAT* in which each sub-model achieves both robustness and diversity by developing the idea of learning an ensemble via adversarial training (AT) and adapt the structure of a recently discovered MIMO architecture. We require the same number of forward and backward passes as AT and yet achieve generalized robustness against multiple attack types. A drawback of our scheme is that if the network size is constrained, it may not have the capacity to accommodate multiple sub-models. Another drawback that is common to most AT methods is that the AE perturbation hyperparameters must be manually selected (we simply use the parameters from MSD [18]). Methods for automatically selecting the best hyperparameters using zero-th order optimization would broaden the applicability of our approach.

## REPRODUCIBILITY STATEMENT

We have publicly released all the code, the datasets and models used, along with the implementation of the attacks. To help with the reproducibility of the experiments, we make available all of our code, the trained models, the training hyperparameter values, and the raw results from our testing. We point to the exact attacks that we have used (which are from existing codebases) in our evaluation. These are all to be found at the anonymized Github link `https://github.com/anonymous-lab-ml/mimo-adv/`. Code and pre-trained models for reproducing our experiments have been uploaded to the anonymized repository `https://github.com/anonymous-lab-ml/mimo-adv/`.

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

## A CODE AND PSEUDO CODE OF *MAT*

### A.1 CODE

Code for reproducing our experiments have been uploaded to the anonymized repository: `https://github.com/anonymous-lab-ml/mimo-adv/`

### A.2 PESUDO CODE

---

**Algorithm 1:** MIMO adversarial training with approximate gradient (inner maximization)

---

**input** : network $f_\theta$, data_size $N$, batch_size $B$, data $x$, label $y^*$, pgd attack steps: $K$,
      num_of_subnetwork: $M$

initialization;

```
/* iterate over all training examples                                        */
```
**for** $i = 0$ *to* `ceil`$(N/B)$ **do**
    $x_{adv} \leftarrow$ `maximize`$(f_\theta, x[i], y[i])$;
    $\theta \leftarrow$ `minimize`$(f_\theta, x_{adv}, y[i])$;
**end**

**def** `maximize` $(f_\theta, x, y^*)$**:**
    $adv = x$ ;                                `//` $adv = (adv_1, \cdots, adv_M)$
    **for** $k = 1$ *to* $K$ **do**
        $y_1, ..., y_M = f_\theta(adv_1, \cdots, adv_M)$;
        $loss = 0$;
        **for** $m = 1$ *to* $M$ **do**
            $loss \leftarrow loss +$ `CELoss`$(y_m, y_m^*)$;
        **end**
        $grad = \nabla_{adv} loss$ ;
        ;                          `// approximate gradient step`
        **for** $m = 1$ *to* $M$ **do**
            $adv_m \leftarrow adv_m + \Pi_{\mathcal{B}_{\|\cdot\|_{p_m}(\epsilon_m)}} (\alpha_m \cdot \mathcal{V}_m(grad))$;
        **end**
    **end**
    **return** $adv$;

**def** `minimize` $(f_\theta, x, y^*)$**:**
    $loss = 0$;
    $y_1, ..., y_M = f_\theta(x_1, \cdots, x_M)$;
    **for** $m = 1$ *to* $M$ **do**
        $loss \leftarrow \max(loss,$ `CELoss`$(y_m, y_m^*))$;
    **end**
    $grad = \nabla_\theta(loss)$;
    $\theta \leftarrow \theta - s \cdot grad$;
    **return** $\theta$

---

## B TIME COMPLEXITY OF DIFFERENT AT METHODS

Compared to standard training (*i.e.,* without ensemble), vanilla AT using $K$-step PGD attack requires $K + 1$ times more computation, due to the $K$ forward and backward passes individually for generating each AE plus the outer minimization. Furthermore, if we try to use $M$-vanilla ensemble AT to promote robustness

over multiple attacks, it will require $M$ times more computation compared to vanilla AT. This becomes infeasible in practice, *e.g.,* for CIFAR-10 with ResNet-50, 100 epochs. It takes 40 times of training time to train one single model compared to standard training. To reduce the computation, we have the insight to replace the vanilla ensemble with a MIMO architecture and to train for all the $M$ attack types within the same network, each within its own subnetwork. This allows us to calculate all subnetworks' outputs $y_i, i = 1, \ldots, M$ in one forward pass instead of $M$ passes, *i.e.,* $(y_1, \ldots, y_M) = f_\theta(x_1, \ldots, x_M)$, instead of $(y_1, \ldots, y_M) = (f_{\theta_1}(x_1), \cdots, f_{\theta_M}(x_M))$ in case of the $M$ vanilla ensemble. This approximation works because we can fit $M$ subnetworks within the original network, *i.e.,* without needing to expand the size of the original network. This is because of the residual capacity in even reasonably sized networks (like ResNet-50 in our case) is sufficient.

Thus, $M$-MIMO AT reduces the number of forward passes to be similar to that of vanilla AT training. However, for each subnetwork, one iteration of PGD attack still needs one backward pass, which leaves the computation of backward pass almost as high as that of naïve ensemble with $M$ networks. The computation cost per iteration of different methods is summarized in Table 6.

Table 6: Computation comparison between different methods (for each batch of data, *i.e.,* each iteration of the outer minimization)

| Methods | Forward passes | Backwards passes |
|---|---|---|
| Standard Training | 1 | 1 |
| Standard MIMO Training [8] | 1 | 1 |
| Vanilla AT ($K$-step PGD) [17] | $K + 1$ | $K + 1$ |
| $M$-Vanilla Ensemble AT ($K$-step PGD) | $M(K + 1)$ | $M(K + 1)$ |
| Multi-perturbation AT ($K$-step PGD) [27] | $MK + M$ | $MK + M$ |
| $M$-MSD AT ($K$-step PGD) [18] | $(1 + M)K + 1$ | $K + 1$ |
| $M$-MIMO AT without Approx. AT($K$-step PGD) | $K + 1$ | $MK + 1$ |
| (Ours) **MAT**: $M$-MIMO Approx. AT ($K$-step PGD) | $K + 1$ | $K + 1$ |
| (Ours) $M$-MIMO MSD ($K$-step PGD) | $(1 + M)K + 1$ | $K + 1$ |

## C    EXPERIMENTS SETTING

### C.1    HYPERPARAMETERS FOR TRAINING THE MODELS

**Models.** We use ResNet18 [9] for all experiments on MNIST dataset. We also reproduce MSD results using the same neural network in their original paper. For CIFAR10, we use ResNet50.

**Optimizers.** For MNIST dataset, we use Adam optimizer [11]. For CIFAR10 dataset, we use SGD optimizier with momentum 0.9, and weight decay $5 \times 10^{-4}$.

**Step size.** We use the step size scheduler follows [25]. For MNIST, the step size linearly increase from 0 to $10^{-3}$ over the first $40\%$ epochs, and down to 0 over the last $60\%$ epochs. For CIFAR10, the step size linearly increase from 0 to 0.1 over the first $40\%$ epochs, and down to 0.005 over the next $40\%$ epochs, and finally back down to 0 in the last $20\%$ epochs. The overall number of epochs is 25 (resp.75) for *MAT* and *MAT +* MSD on MNIST (resp. CIFAR10) datasets.

**Hyperparameters for adversarial training.** For all MNIST dataset, the inner maximization is optimized by $(\ell_\infty, \ell_2, \ell_1)$ PGD attacks with max iteration 100, radius $\epsilon = (0.3, 2.0, 10)$ and step size $\alpha = (0.01, 0.1, 0.8)$ respectively. For all CIFAR10 dataset, the inner maximization is optimized by $(\ell_\infty, \ell_2, \ell_1)$ PGD attacks with max iteration 50, attack radius $\epsilon = (0.03, 0.5, 12)$ and step size $\alpha = (0.003, 0.02, 1.0)$ respectively.

### C.2 HYPERPARAMETERS FOR ADVERSARIAL ATTACKS USED FOR ROBUST TEST

For all the results, we evaluate the first 1000 test examples. To verify the performance of *MAT* and compare it to the performance of the baseline Adversarial Training, we evaluate the performance of our model against $\ell_\infty, \ell_2, \ell_1$ PGD attacks [17], Carlini Wagner [4] $\ell_2$ attack,and two decision-based black-box attack: Boundary attack [2] and HopSkipJump attack[5].

For PGD attacks, to guarantee the convergence, we increase the max iteration to 200. For $\ell_1$ PGD attack, we fixed the number of pixels to perturb per iteration as 20 during test. The other hyperparameters stay the same as in the software distribution.

For other attacks, we set the hyperparameters to be consistent to the original works.

## D THE DIFFERENCE BETWEEN THE TRUE GRADIENT AND APPROXIMATE GRADIENT AFTER THE STEEPEST DIRECTION MAPPING OPERATOR $\mathcal{V}$

We claim that the cross gradient does influence the optimization, since the the small value of cross gradient may affect the gradient value after the steepest descent direction. In Figure 4, we define $E_p = \frac{\sum(\mathcal{V}_p(\nabla_{x_m}(\mathcal{L}_1 + \mathcal{L}_2 + \mathcal{L}_3)) \oplus \mathcal{V}_p(\nabla_{x_m}\mathcal{L}_m))}{|x_m|}$, where $|\cdot|$ denotes the cardinality of the $x_m$, $\oplus$ denotes XOR operation, and $p = \{1, \infty\}$. $E_p$ calculates the fraction of entries that are different in the approximate gradient *after* steepest-descent projection compared to the true gradient. Notice that the XOR operator with a summation calculates the overall number of different pixels the two gradients have.$|x_m|$ denotes the overall pixels the gradient has. Figure 4 shows that the difference between the approximate gradient and the true gradient after the steep ascent direction mapping, where it decreases during training. Compared to Figure 1, which shows the difference between the true gradient and our approximation. The y axis denotes the fraction of pixels that changed by the cross gradient term. We can see from 4, especially, $E_\infty$ does not converge to 0. They term might perform an important role to bring multiple robustness for the sub-model trained against a single attacks.

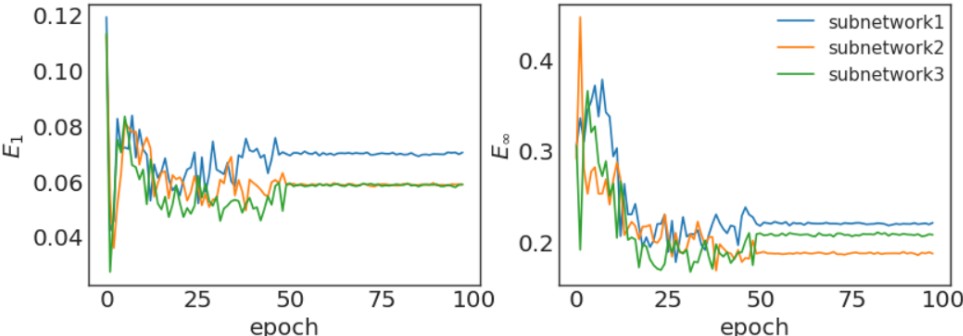

Figure 4: Approximate gradient difference caused by steepest descent operator for $\ell_1, \ell_\infty$ norms: epoch versus the disagreement $E$.

## E  WALL CLOCK TIME DURING TRAINING

In this section we report the wall clock time to train the different models.

Table 7: The wall clock time for training the model. All the models are training using ResNet-18 (resp. ResNet-50) on MNIST (resp.CIFAR10).

|  | Standard Training | AT | MSD | MAT |
|---|---|---|---|---|
| Training time on MNIST | 10min23s | 8h12min | 12h56min | 10h02min |
| Training time on CIFAR10 | 40min11s | 89h42min | 141h23min | 123h07min |

## F  SUB-MODEL TRAINED WITH SAME ADVERSARIAL ATTACK

As mentioned, *MAT* can also be integrated with other AT techniques to strengthen the model. In this section, we empirically show the three experiments where we combine *MAT* with three previous methods, $\ell_\infty$ AT, $\ell_2$ AT, and MSD respectively. The result in Table 8 shows that *MAT* achieves higher robustness accuracy in most scenarios even against the attack type that AT approach has been designed for. . Notice that all these models are trained with the same models and hyperparameters. As we observe, for MSD+MAT, we also bring $20.4\%$ extra robustness against $\ell_\infty$ attacks, and $29.2\%$ more robustness accuracy against $\ell_1$ attacks with $0.8\%$ extra clean accuracy. The drawback is we trade off $27\%$ robust accuracy on $\ell_2$ attacks. This is due to the imbalance of choices of the hyperparameters giving relative weight to the three attack types $\ell_1$, $\ell_2$, and $\ell_\infty$. In this table, we show our MAT could use with sub-models trained with same metrics. The results show that if we combine the MIMO structures and $\ell_\infty$ AT, we could achieves overall higher accuracy comparing to the standard $\ell_\infty$ AT.

## G  TRANSFERABILITY AND SUB-MODEL ROBUSTNESS

This section shows the sub-models performance of the vanilla ensemble (resp. *MAT*) against $\ell_2$ PGD attacks in Table 9 (resp. Table 10) and against $\ell_1$ PGD attacks in Table 11 (resp. Table 12). Comparing Table 9 and Table 10, when the attacker generates the adversarial examples on sub-model 2, we find that the adversarial examples is easier to transfer to other sub-models comparing to our *MAT*. This especially is due to the vulnerability of the sub-model 1 and sub-model 3. Since the sub-models is not robust, if we check the second row of the vanilla-ensemble model, the number of successful $\ell_2$-attacks is 437 which is much higher than 383 of our *MAT*. Defending against $\ell_1$ attacks, though the sub-model of the vanilla ensemble achieves low adversarial transferability, our model still outperforms the vanilla ensemble by $12.1\%$. This is due toe the vulnerable of the sub-models, which empirically proves our claim that to achieve robustness against adversarial examples, both sub-model's robustness and the adversarial diversity between sub-models are required.

Table 8: Three sub-models trained with same Adversarial Attacks on MNIST: the first row shows the clean accuracy; the remaining rows corresponds to different attacks; different column compares the robustness accuracy of $\ell_\infty$ AT (resp. $\ell_2$ AT or MSD) with MAT($\ell_\infty$) (resp. MAT($\ell_2$) or MAT+MSD), where MAT($\ell_\infty$) (resp. MAT($\ell_2$)) denotes MAT with each sub model trained using $\ell_\infty$-attack (resp. $\ell_2$-attack), and MAT+MSD denotes MAT with each sub model trained using MSD.

| | $\ell_\infty$ AT | MAT($\ell_\infty$) | | $\ell_2$ AT | MAT($\ell_2$) |
|---|---|---|---|---|---|
| clean accuracy | **99.3**% | 99.2% | clean accuracy | **98.6**% | 98.5% |
| PGD-$\ell_\infty$ Attack | 96.0% | **96.1**% | PGD-$\ell_2$ Attack | **76.8**% | 76.4% |
| HSJ Attack | **98.0**% | 97.8% | C & W Attack | 76.8% | **77.2**% |
| AutoAttack | 94.9% | **95.6**% | Boundary Attack | 78.2% | **79.0**% |
| All $\ell_\infty$ Attacks | 94.9% | **95.6**% | All $\ell_2$ Attacks | 74.4% | **75.7**% |

MAT($\ell_\infty$)                                        MAT($\ell_2$)

| | MSD | MAT+MSD |
|---|---|---|
| clean accuracy | 98.5% | **99.3**% |
| PGD-$\ell_\infty$ Attack | 62.2% | **82.4**% |
| HSJ Attack | 87.4% | **97.3**% |
| All $\ell_\infty$ Attacks | 62.2% | **82.4**% |
| PGD-$\ell_2$ Attack | 72.0% | **77.7**% |
| C & W Attack | **69.8**% | 62.9% |
| Boundary Attack | **70.6**% | 55.0% |
| All $\ell_2$ Attacks | **67.3**% | 39.7% |
| PGD-$\ell_1$ Attack | 63.7% | **93.7**% |
| Salt & Pepper Attack | 84.5% | **94.9**% |
| All $\ell_1$ Attacks | 62.7% | **91.9**% |
| All Attacks | **54.7**% | 39.5% |

MAT + MSD

Table 9: Vanilla-Ensemble diversity testing: successful $\ell_2$-adversarial attack. Each column shows the misclassification samples out of 1000. The first (resp. second) row corresponds to $\ell_2$ adversarial attacks generated from model 2 (resp. vanilla-ensemble). The last row relates to clean accuracy with no attack.

|  | Sub-model 1 | Sub-model 2 | Sub-model 3 | Vanilla-Ensemble |
| --- | --- | --- | --- | --- |
| Sub-model 2 | 336(145.5%) | 231 | 298(129.0%) | 234(101.3%) |
| Vanilla-Ensemble | 557 | 206 | 704 | 437 |
| No Attack | 14 | 9 | 12 | 12 |

Table 10: *MAT* diversity testing: successful $\ell_2$ adversarial attack. Each column shows the misclassification samples out of 1000. The first (resp. second) row corresponds to $\ell_\infty$ adversarial attacks generated from sub-model 2 (resp. *MAT*). The last row relates to clean accuracy with no attack.

|  | Sub-model 1 | Sub-model 2 | Sub-model 3 | MAT |
| --- | --- | --- | --- | --- |
| Sub-model 2 | 251(76.8%) | 327 | 260(79.5%) | 272(83.2%) |
| MAT | 324 | 284 | 363 | 320 |
| No Attack | 9 | 7 | 7 | 5 |

Table 11: Vanilla-Ensemble diversity testing: successful $\ell_1$-adversarial attack. Each column shows the misclassification samples out of 1000. The first (resp. second) row corresponds to $\ell_\infty$ adversarial attacks generated from model 2 (resp. vanilla-ensemble). The last row relates to clean accuracy with no attack.

|  | Sub-model 1 | Sub-model 2 | Sub-model 3 | Vanilla-Ensemble |
| --- | --- | --- | --- | --- |
| Sub-model 3 | 100(16.4%) | 64(10.5%) | 608 | 141(23.2%) |
| Vanilla-Ensemble | 413 | 188 | 556 | 383 |
| No Attack | 14 | 9 | 12 | 12 |

Table 12: *MAT* diversity testing: successful $\ell_1$ adversarial attack. Each column shows the misclassification samples out of 1000. The first (resp. second) row corresponds to $\ell_\infty$ adversarial attacks generated from sub-model 2 (resp. *MAT*). The last row relates to clean accuracy with no attack.

|  | Sub-model 1 | Sub-model 2 | Sub-model 3 | MAT |
| --- | --- | --- | --- | --- |
| Sub-model 3 | 79(45.7%) | 54(31.2%) | 173 | 81(46.8%) |
| MAT | 273 | 249 | 287 | 262 |
| No Attack | 9 | 7 | 7 | 5 |

