# OpenReview forum: "Resilience to Multiple Attacks via Adversarially Trained MIMO Ensembles"
_ICLR.cc/2022/Conference — ICLR 2022 Submitted_

### Official Review · Reviewer_U6VC · 2021-11-01

**Correctness:** 3
**Technical Novelty And Significance:** 3
**Empirical Novelty And Significance:** 2
**Recommendation:** 3
**Confidence:** 4

**Main Review:**

Pros:

(1) The motivation behind is simple and interesting.

(2) The problem of defending multiple perturbations is significant.

(3) The insights given by the author are interesting for me.

Cons:

(1) My biggest concern for this paper is the experiment. First of all, the experimental settings are ambiguous. For example, what is the perturbation budget for your experiment? What is the step size of PGD attack? Did the author clip the perturbation for the C&W attack? Why didn't the author use AutoAttack for the main experiment on CIFAR-10 and MNIST? It seems that the author missed these important experimental settings, which make the results unconvincing.

(2) The author missed some studies that also aim to improve model robustness against multiple adversarial perturbations [R1]. The author should compare and discuss with them.

(3) According to the experimental results in Table 6, MAT+MSD achieves comparable performance with MSD, and even shows weaker robustness on the "All Attacks". How did the author explain this phenomenon?

(4) Again, as shown in Section 4.4, the author claimed that ''The drawback is we trade-off 27% robust accuracy on $\ell_2$ attacks. This is due to the imbalance of choices of the hyperparameters''. This is not convincing to me, and the discussion is somewhat tricky. In other words, if the author fine-tune the hyperparameter to increase the robustness against $\ell_2$ attacks, would the model show weak robustness against other types of attacks?

(5) Moreover, I also found the ''sub 2'' model only achieves 33.1\% on $\ell_{\infty}$ PGD attacks, while other sub-networks achieve 80+\% robustness on $\ell_{\infty}$ PGD. Could the author provide more explanations?

(6) What about the defense performance on unseen perturbations? For example, training on $\ell_1$ and $\ell_2$ adversarial perturbations and evaluating the robustness on $\ell_{\infty}$ adversarial attacks.

References

[R1] Towards Defending Multiple Adversarial Perturbations via Gated Batch Normalization, arxiv, 2020.

**Summary Of The Paper:**

This paper proposes an ensemble based adversarial training strategy, which could improve the worst-case model robustness against multiple $\ell_p$-norm adversarial perturbations. To improve the training efficiency, the author introduces the Multiple-Input Multiple-Output (MIMO) strategy. The motivations and insights provided by this paper are interesting for me.

**Summary Of The Review:**

The proposed method is very interesting and the motivation behind is intuitive. However, I think the experiments are not enough to prove its effectiveness, and some settings are even problematic.

---

> ### Author Response · Authors · 2021-11-23
> **Response to reviewer U6VC (Part 1)**
>
> **My biggest concern for this paper is the experiment. First of all, the experimental settings are ambiguous. For example, what is the perturbation budget for your experiment? What is the step size of PGD attack? Did the author clip the perturbation for the C\&W attack? Why didn't the author use AutoAttack for the main experiment on CIFAR-10 and MNIST? It seems that the author missed these important experimental settings, which make the results unconvincing.**
>
> Thanks for the comments. Four points need to clarify here:
>
> 1. The perturbations are equal adopting the same setting as in MIMO (please see Sec. 2.3 for detailed setting for MIMO). This is a natural assumption since the adversarial attacks are only allowed to perturb the inputs but not the model structure.  During testing, the same inputs are copied and fed to all the $M$ sub-models (notice copy procedure is a part of the model structure), then MAT decides the prediction by averaging $y_{i,m}$. In the real-world setting, it is also natural to only expose the interface for the users to upload a single batch of images.
>
> 2. The perturbation we report in table 4 is $(0.3,2,10)$ for $\ell_\infty,\ell_2,\ell_1$ attacks with step size $\alpha=(0.01,0.1,0.8)$ on MNIST iterating $100$ steps, and the perturbation in table 5 is $(0.03,0.5,12)$ for $\ell_\infty,\ell_2,\ell_1$ attacks with step size $\alpha=(0.003,0.02,1.0)$ on CIFAR10 iterating $50$ steps. [1,2]
>
> 3. For C\&W attack and other attacks seeking for the minimum perturbation like Boundary Attacks, etc. we only count as success when a) the adversarial example successfully fools the model, and b) the perturbation is inside the $\epsilon$ ball;
>
> 4. We include the AutoAttack results in the main results tables, Table 4 and Table 5 now. Please check appendix C for the detailed experimental setting.
>
> **The author missed some studies that also aim to improve model robustness against multiple adversarial perturbations [R1]. The author should compare and discuss with them.**
>
> Thanks for pointing this paper. It is a really nice work that also consider the multiple robustness. The mean idea of this paper is to create a single neural network with Gated Batch Normalization (GBN) to improve robustness, where GBN consists of a gated subnetwork and a multi-branch batch normalization (BN) layer. Its neural network structure also a multi-out structure. A study on this work might be intersting.
> However, given the relatively recent release of this paper on arXiv, we were unable to empirically compare to this paper.
>
> **According to the experimental results in Table 6, MAT+MSD achieves comparable performance with MSD, and even shows weaker robustness on the "All Attacks". How did the author explain this phenomenon?**
>
> MAT+MSD is a experiment exploring the possibility to combine the previous MSD method with MAT frameworks. Our experiments show that MAT+MSD is harder to converge comparing to MAT. Further study on how MAT combined different previous robustness methods is also worth exploring in future work.
>
> **Again, as shown in Section 4.4, the author claimed that "The drawback is we trade-off $27\%$ robust accuracy on $\ell_2$ attacks. This is due to the imbalance of choices of the hyperparameters". This is not convincing to me, and the discussion is somewhat tricky. In other words, if the author fine-tune the hyperparameter to increase the robustness against attacks, would the model show weak robustness against other types of attacks?**
>
> We adopted the same hyperparameters as the MSD method, including the step size of the PGD attack, attacks radius $\epsilon,$ and termination max iteration. That is, all the hyperparameters are chosen as the best tuning for MSD in order to present results on the same baseline (which may not be the best for MAT). Please check Sec. C1. for detailed information about hyperparameter tuning. Focusing on CIFAR-10, we agree with the reviewer that the MSD method seems to provide better performance on  $\ell_2$- PGD attacks as the test radius is less than $0.5$. However,  MAT outperforms MSD whenever the radius is larger than $0.5$  (See Figure. 3 for support).  We plot the test accuracy curves of MAT+MSD, MAT, and MSD on MNIST under PGD $\ell_\infty$, $\ell_2$, and $\ell_1$ attacks with varying attack radius on CIFAR10 dataset in Sec. 4.2, please see Figure. 3 for further information.

---

> ### Author Response · Authors · 2021-11-23
> **Response to reviewer U6VC (Part 2)**
>
> **Moreover, I also found the "sub 2" model only achieves $33.1\%$ on $\ell_\infty$ PGD attacks, while other sub-networks achieve $80+\%$ robustness on $\ell_\infty$ PGD. Could the author provide more explanations?**
>
> Thanks for the comments. We found that this part is not easy to follow, so we re-wrote section 4.1 to make it clear. Notice that our method aims for the overall robustness against multiple kinds of adversarial attacks. We want to clarify this in two points. 1) It is not beyond our expectation that the sub-models performance against one specific kind of adversarial attack may not be good. Our table 2 of the revised paper shows that when the attacker tries to use $\ell_\infty$ PGD attack to fool our MAT, $548$ out of $1000$ adversarial examples fools the sub-model 2. However, our MAT only misclassifies $128$ out of these $1000$ adversarial examples. The performance of our MAT is close to the two best performing sub-models. 2) This sub-model 2 achieves the best robustness against $\ell_2$ and $\ell_1$ PGD attacks (See Table. 11 and Table. 12 in the appendix G of the revised paper for the detailed results). This means that the sub-models of MAT all contribute useful outputs to the final results.
>
> **What about the defense performance on unseen perturbations? For example, training on $\ell_1$ and $\ell_2$ adversarial perturbations and evaluating the robustness on $\ell_\infty$ adversarial attacks.**
>
> Thanks for the comments. It is an interesting question to explore. However, training using $\ell_1$ and $\ell_2$ does not bring robustness against $\ell_\infty$ attacks because of the geometry in high dimensions [2]; the $\ell_\infty$ norm ball is significantly different than the $\ell_1$ and $\ell_2$ norm balls. In particular, for the standard $\epsilon$'s used, the $\ell_{\infty}$-bounded ball can be significantly larger than the other two.
>
> [1] Maini, Pratyush, Eric Wong, and Zico Kolter. "Adversarial robustness against the union of multiple perturbation models." International Conference on Machine Learning. PMLR, 2020.
>
> [2] Tramer, Florian, and Dan Boneh. "Adversarial training and robustness for multiple perturbations." arXiv preprint arXiv:1904.13000 (2019).

---

### Official Review · Reviewer_YRoM · 2021-11-01

**Correctness:** 3
**Technical Novelty And Significance:** 2
**Empirical Novelty And Significance:** Not applicable
**Recommendation:** 5
**Confidence:** 4

**Main Review:**

[Strength]

1 To defend against multiple attacks, the proposal MAT requires that a) each sub-model is robust against a specific attack, and b) submodels have reduced transferability.

2 Compared with vanilla AT+ensemble, the proposal MAT is computationally efficient.

[Weakness]

1 In my opinion, this paper's novelty is not strong, e.g., sub-models should be diverse, sub-models are adversarially robust, and MIMO training strategy.

2 Sections 3.1 and 3.2 is not clearly written. E.g., what does V in equation 5 mean?

3 Compared with MSD (Maini et at.), the proposal MAT has marginal improvements against multiple attacks.

[Questions]

1 Could authors illustrate more about Eq. 6 and Eq. 7. From the current writing, I do not understand cross gradient in detail.

2 A.2 PSEUDO CODE, where does penalty of cross gradient apply?

**Summary Of The Paper:**

[Summary]
This paper proposes an ensemble method to defend against adversarial attacks.
To be specific, this paper combines MIMO strategies (Havasi et al.) and adversarial training (Madry et al.).

**Summary Of The Review:**

My main concern is the novelty of this paper;
Besides, this paper is not well written.



##### Post rebuttal ####

Many thanks for authors' feedback.
I have read other reviewers' comments and corresponding feedback.
I agree with other reviewers' evaluations such as “limited novelty“, "weak experimental evidence", etc.
Thus, I keep my score unchanged.

---

> ### Author Response · Authors · 2021-11-23
> **Response to reviewer YRoM**
>
> **Compared with MSD (Maini et at.), the proposal MAT has marginal improvements against multiple attacks.**
>
> We’d like to stress two points here.
> 1. Our work shows that the ensemble method could improve the adversarial robustness in the real-world setting. Prior work either failed, or mainly focused on the theoretical analysis, which only provides robustness with a very small radius.
> 2. As we have mentioned in the paper, MAT could be useful because it is competitive but more computationally efficient than MSD, that is, $K+1$ forward passes for MAT versus $(1+M)K+1$ for MSD, with the same number of backward passes $K+1$. 3) We add some new results in Sec. 4.2.
>
> **Could authors illustrate more about Eq. 6 and Eq. 7. From the current writing, I do not understand cross gradient in detail.**
>
> At a high-level, the true gradient for sub-model $i$ would only be with respect to loss for sub-model $i$. However, we compute the gradient with respect to the *sum* of the sub-model losses. The cross-gradient is merely the difference between these two gradients. The use of the gradient with respect to the *sum* of sub-model losses helps to . We include the pseudo-code of MAT in appendix Sec. A.1, where the cross loss related to cross gradient is used which allow the exchange of information between the networks during the attack stage which may help in reducing the transferability of AEs during the training. It plays an essential role in improving the adversarial diversity of MAT and reducing the computation burden.
>
> **A.2 PSEUDO CODE, where does penalty of cross gradient apply?**
>
> Thanks for the comment. In the pseudo-code of our model in appendix Sec. A.2, where in the comment "approximate gradient step", for the input $x_i$ of each sub-model $m$, at each iteration, the gradient is calculated with respect to the summation of the loss. (in the pseudo-code, the loss $\mathcal{L}$ is chosen as cross entropy loss "CELoss", we will add this information in the revised version) rather than its true loss. Taking derivative of the summation loss naturally includes the cross gradient term which allow the exchange of information between the networks during training. Because we used the approximate/inexact gradient instead of the true gradient of sub-model $m$ with data batch index $i.$

---

### Official Review · Reviewer_zrcs · 2021-11-02

**Correctness:** 4
**Technical Novelty And Significance:** 3
**Empirical Novelty And Significance:** 2
**Recommendation:** 5
**Confidence:** 4

**Main Review:**

Strengths:
1. The algorithm is intuitive and the claims are well supported by experiments and analysis.
2. The idea of cross gradients as a metric to analyse effect of transferable perturbations due to the unique architecture of MIMO models is interesting and worth further study on its own.
3. The paper is organized well and the ideas are presented in a coherent manner.

Weaknesses:
1. MAT generally seems to perform worse than MSD for all three threat models when compared head to head. The robust accuracy for $\ell_2$ attacks is significantly lower when compared to both MSD and AT. The paper claims in another section that this is due to choice of hyperparameters. Could the authors clarify this?
2. It is not clear from the text what the threat model or the attack budget is.  Are the various copies of the input image allowed individual perturbations or are the perturbations forced to be equal?
3. The authors do not test their defense against an adaptive attack. For example, an attacker could try to find salient sub-models using loss values for each input and perturb just their inputs. This also ties into my second question regarding the clarity of the threat model.
4. As the authors claim robustness against multiple attacks, ideally robust accuracy should be reported for the worst case attack in the combined threat model for a more clear picture.
5. The authors don't seem to have used the cross gradient terms as a penalty during training. Is there any specific reason for that?

**Summary Of The Paper:**

In this work, the authors propose a new training algorithm MAT that adversarially trains Multi Input Multi Output (MIMO) models. They show that ensemble models based on MIMO when trained adversarially , show ``adversarial diversity'' and therefore are less vulnerable to transfer attacks. They empirically demonstrate that such models are robust to a variety of $\ell_1, \ell_\infty$ and $\ell_2$ attacks and achieve better performance than other methods claiming robustness across different threat models. The authors also show computational benefits of their algorithm over vanilla ensemble training.

**Summary Of The Review:**

The authors propose an algorithm for adversarial training of MIMO models. They claim better robustness against a wide variety of threat models, and demonstrate this with experiments on CIFAR-10 and MNIST. The claims are mostly well supported, however the improvements seem marginal.The technical contribution is therefore somewhat limited. The authors also need to clearly specify the threat model that they assume for their experiments. However, their results for the cross-gradient  across the sub-models is an interesting addition and does show that their method holds promise. Reviewing all of this, I currently this paper needs some work, and therefore recommend a marginal rejection.

---

> ### Author Response · Authors · 2021-11-23
> **Response to reviewer zrcs (Part 1)**
>
> **MAT generally seems to perform worse than MSD for all three threat models when compared head to head. The robust accuracy for $\ell_2$ attacks is significantly lower when compared to both MSD and AT. The paper claims in another section that this is due to choice of hyperparameters. Could the authors clarify this?**
>
> Thanks for the comments.
> There are three sub-parts to this question. (1) Does MAT generally seem to perform worse than MSD for all three threat models when compared head to head? (2) Is the robust accuracy of \name for $\ell_2$ attack significantly lower when compared to both MSD and AT? and (3) why may MAT need further hyperparametertuning?
>
> 1. It depends on the dataset. The performance of MAT is comparable w.r.t MSD on CIFAR-10 with attack radius  $0.03$ (resp. $0.5$ and $12$) for $\ell_\infty$ (resp. $\ell_2$ and $\ell_1$);  on MNIST, MAT outperforms MSD on all $\ell_{\infty}$ attacks by $82.3\%-56.3\%=26\%,$ MAT is worse than MSD by $10\%$ (resp. $3\%$) on $\ell_{2}$ (resp. $\ell_1$). Note that as we stressed in the paper, MAT may be attractive because it is competitive but more computationally  efficient than MSD, that is, $K+1$ forward passes for MAT versus $(1+M)K+1$ for MSD, with the same number of backward passes $K+1$.
>
> 2. It depends on the attack radius and data set. For MNIST, on $\ell_{2}$-PGD attack,  MAT+MSD surpass the other MAT or MSD alone. For CIFAR-10, MSD exhibits better testing accuracy than the other two methods on CIFAR-10 with radius $r=0.5$, however, MAT outperforms MSD when the attack radius increases (greater than the default choice $0.5$). We add the robust accuracy with varying radius in the revised version. Please see Sec. 4.2, Figures 2 and 3 for more details.
>
> 3. We adopted the same hyperparameters as the MSD method, including the step size of the PGD attack, attack radius $\epsilon,$ and termination max iteration. That is, all the hyperparameters are chosen as the best tuning for MSD in order to present results on the same baseline (which may not be the best for MAT). Please check Sec. C1. for detailed information about hyperparameter tuning. Focusing on CIFAR-10, we agree with the reviewer that the MSD method achieves higher accuracy on  $\ell_2$- PGD attacks with the test radius greater $0.5$. However,  MAT outperforms MSD whenever the radius is larger than $0.5$  (See Figure 3 for support).  We plot the test accuracy curves of MAT+MSD, MAT and MSD on MNIST under PGD $\ell_\infty$, $\ell_2$, and $\ell_1$ attacks with varying attack radius on CIFAR10 dataset in Sec. 4.2, please see Figure. 3 for further information.
>
> **It is not clear from the text what the threat model or the attack budget is. Are the various copies of the input image allowed individual perturbations or are the perturbations forced to be equal?**
>
> Thanks for the comment, we will add this information in Sec. 3.
> The perturbations are equal adopting the same setting as in MIMO (please see Sec. 2.3 for detailed setting for MIMO) in which, at test time, the model copies the each input for all 3 submodels, then the model decides the prediction by averaging the predictions $y_{i,m}$.
> In the real world setting, it is also natural to only expose the interface for the users to upload a single image. Thus, the perturbations are the same for each of the 3 submodels just as in MIMO all 3 images are the same at test time.
>
> **The authors do not test their defense against an adaptive attack. For example, an attacker could try to find salient sub-models using loss values for each input and perturb just their inputs. This also ties into my second question regarding the clarity of the threat model.**
>
> We'd like to stress here that all the white-box attacks are adaptive attacks against the adversarial training model. We adopt the same setting as in the literature of adversarial training [1,2,3,4,5,6]. Notice that the difference between adaptive attacks
> and white-box attacks is whether the attackers are aware of the defense, in our case, the model is fixed after training, therefore, the attackers in the white-box setting already have full knowledge of our defense. That means in our setting, the adaptive attacks are the same as the white-box attacks. All the evaluations are based on these attacks.
>
> **As the authors claim robustness against multiple attacks, ideally robust accuracy should be reported for the worst case attack in the combined threat model for a more clear picture.**
>
> We reported the worst-case attack in the combined threat model in the main result. Please check the row called ``all attack'' in Table. 4 and Table. 5.  To make it more clear, we will add more explanation on the worst-case attack as the reviewer suggested, please check the revised version Sec. 4.3. Thank you for the suggestion.

---

> ### Author Response · Authors · 2021-11-23
> **Response to reviewer zrcs (Part 2)**
>
> **The authors don't seem to have used the cross gradient terms as a penalty during training. Is there any specific reason for that?**
>
> Thanks for the comment.
> We'd like to clarify that we used the cross gradient terms which allow the exchange of information between the networks during the attack stage which may help in reducing the transferability of AEs during training.  We defined the cross gradient as equation (7) in Sec. 3.2. During the training, we used the approximate gradient instead of the true gradient. The cross gradient terms is an extra term of the loss function during training. We also included the pseudo-code of our model in appendix Sec. A.1, where in the comment ``approximate gradient step'', for the input $x_i$ of each sub-model $j$, at each iteration, the gradient is calculated with respect to the summation of the loss rather than its true loss. Taking derivative of the summation loss naturally includes the cross gradient term during the training.
>
> [1] Madry, Aleksander, et al. "Towards deep learning models resistant to adversarial attacks." arXiv preprint arXiv:1706.06083 (2017).
>
> [2] Maini, Pratyush, Eric Wong, and Zico Kolter. "Adversarial robustness against the union of multiple perturbation models." International Conference on Machine Learning. PMLR, 2020.
>
> [3] Tramer, Florian, and Dan Boneh. "Adversarial training and robustness for multiple perturbations." arXiv preprint arXiv:1904.13000 (2019).
>
> [4] Zhang, Hongyang, et al. "Theoretically principled trade-off between robustness and accuracy." International Conference on Machine Learning. PMLR, 2019.
>
> [5] Wong, Eric, Leslie Rice, and J. Zico Kolter. "Fast is better than free: Revisiting adversarial training." arXiv preprint arXiv:2001.03994 (2020).
>
> [6] Shafahi, Ali, et al. "Adversarial training for free!." arXiv preprint arXiv:1904.12843 (2019).

---

> ### Comment · Reviewer_zrcs · 2021-11-24
> **Thanks for the clarifications**
>
> I thank the authors for the detailed response and further addressing my concerns and questions. However, the revisions seem to show that MAT is just about comparable or worse than MSD on its own (on CIFAR-10, fig. 3a, b). And the methods together surprisingly seem to do worse. The MNIST results are a lot more promising but given that MNIST is considered a toy dataset by most recent work, it is difficult to accept the results as being generally valid. Fig 3b also seems to suggest that stronger attacks will break MAT models more easily. However, I appreciate the authors for clearly demonstrating computational advantages in Appendix Sec. E., which does make MAT an interesting option in low-resource settings.
>
> I also disagree that the current white-box attacks are the only possible adaptive attack. The MIMO branches are trained with the assumption that they will pick out varying features. An adaptive attack here would have to take that into consideration and induce varying adversarial noises for each branch, perhaps enforcing the epsilon budget on the sum of the adversarial noise.
>
> Due to these concerns, I am inclined to stick with my earlier review. I welcome any additional discussion from the authors.

---

> > ### Author Response · Authors · 2021-11-30
> > **Response to reviewer zrcs**
> >
> > Thanks for followup comments. There seem to be three concerns which we address separately: 1) MAT+MSD does not perform as well on CIFAR10. 2) Fig 3b suggests that a stronger attacks might work against MAT. 3) There exists adaptive attacks that attack all 3 submodels.
> >
> > 1. Our main contribution is MAT, and we mainly focus on the performance of MAT itself. MAT+MSD sometimes outperforms MAT and MSD(fig 2a $\ell_2$ PGD attack) which brings the possibility that MAT with other methods could bring us stronger models. We conjecture that the main reason MAT+MSD does not always perform better is because it is a harder optimization problem to solve and does not converge as easily.
> >
> > 2. Fig 3b shows that MAT models either performs comparably (against $\ell_\infty$ and $\ell_1$ attacks) or outperforms MSD (against $\ell_2$) when the perturbation is large.
> >
> > 3. Thanks for your suggestion on an adaptive attack. If we understand correctly, you are suggesting attacking each sub-model separately (using perhaps 1/3 of the epsilon for each sub-model). Is that correct? If so, this is like doing 3 independent attacks against the sum/avg of the outputs. Why would this be better than optimizing the sum/avg directly as we do in our experiments? Usually, global optimization (e.g., over the sum/avg) performs better than local optimization (e.g., over each sub-model individually). Though maybe we are missing something here. More generally, to perform an adaptive attack, we first need to build an objective function, which contains two parts. The first part tries to flip the classification where the most effective way is to attack the sum/avg of the outputs. The second part tries to control the overall perturbation. Currently, if we only use one metric like CW attacks. The results already show that the CW attack does not succeed with high probability. With more constraints like EAD which brings an extra $\ell_1$ penalty term, the optimal value for maximizing the classification error term will not increase.

---

### Official Review · Reviewer_7nra · 2021-11-03

**Correctness:** 3
**Technical Novelty And Significance:** 2
**Empirical Novelty And Significance:** 2
**Recommendation:** 5
**Confidence:** 5

**Main Review:**

# Strengths
- This paper focuses on the important problem of robustness to adversarial attacks using a straightforward idea that adds computationally negligible cost at inference time, which could be of interest to the community. Using MIMO architectures for AT is interesting.
- It presents extensive experiments on MNIST, including varying ball sizes (fig. 2), and a comparison to the adaptive AutoAttack.

----

# Weaknesses
Given that the paper combines existing work (MIMO) and AT, the novelty in my opinion is limited. Moreover, the contributions are empirical. This raises the need for a more thorough empirical analysis to demonstrate the efficacy of the proposed approach.

- (Major) Weak empirical evaluation.
   - While the authors compare with existing methods on CIFAR-10, comparison on other real-world datasets is missing, given that the paper is empirical.
   - While the authors perform a thorough analysis on MNIST, to argue in favor of MAT it would be helpful to have such analysis on CIFAR-10 as MNIST can be in some cases misleading. For example, the paper lacks results on CIFAR-10 with: (1) varying radius,
   - The paper lacks experiments with a varying number of attack steps, in particular, showing its generalization and robustness for different a varying number of steps.
  - Solely two architectures are used, thus it is unclear if MAT performs well using different models.

- (Major) Unclear if the MAT method is outperforming existing ones.
   - Focusing on CIFAR-10, the MSD method [20] seems to provide better performances across the different attacks, see Tab.5.  The authors acknowledge this and argue that MAT may need further hyperparameter tuning, which raises a concern about how the hyperparameters are selected, as ideally the selection of the presented results should be fair/fixed toward all methods.
   - If I am correct, from Tab. 9 in the App.  MAT without $\ell_\infty$ is less robust to AutoAttack relative to MSD.
- (Minor) While the flow is easy to follow, the writing is in some cases informal and thus ambiguous, e.g., in the abstract 'attacks do not transfer easily', and careful proofreading and polishing are needed.
- (Minor) In the introduction the authors argue with certainty that ensemble methods fail due to two main reasons, there should be a citation, or alternatively, elaborate that this is argued in this paper and point to the relevant section.  See also the relevant question 4 below.

----
# Minor comments and Recommendations
- It would be helpful to include the total training times per epoch for the different methods for fixed setup.
- Include detailed captions, e.g. dataset for each table, etc. Similarly for the tables, rather 'main result on...'  be more precise, e.g. rows/columns are methods used for training or testing, etc.
- Incomplete sentence in App. C.1
- Sec 4.3: Our methods also increasing -> increase

----
# Questions
1. Resnet-50 for CIFAR-10 is less-common, did the authors try with ResNet-18? Moreover, what are the models used by MSD [20]? Ideally, the authors should include a comparison between the two methods using the same model for a fair comparison.
2. For how many epochs are MAT and the baselines trained?
3. I would recommend moving the AutoAttack results on CIFAR-10 in the main paper, as such comparison is important. Moreover, how do these AutoAtttack results compare to state-of-the-art AT methods (other than MSD)?
4. Looking solely at Tab.1 could also indicate that the two AT models are robust. What are the clean accuracies and PGD robustness of the models in Tab.1 both separately and the two as an ensemble?
5. Did the authors compare against PGD-50-10?




**Summary Of The Paper:**

This paper uses ensembles--and more precisely multi-input-multi-output (MIMO) neural networks for computational efficiency---for adversarial training (AT), resulting in a method called MAT, short for MIMO AT.
Moreover, the authors investigate how the sub-networks should be trained to increase the overall robustness of the total MIMO model.
In particular, the proposed MAT generates the adversarial samples using the gradient with respect to the objective of the ensemble.

The authors demonstrate on MNIST and CIFAR-10 that the proposed method achieves generalization and robustness comparable to existing state-of-the-art methods.


**Summary Of The Review:**

In summary, the proposed method is intuitive and simple, as well as computationally attractive.
However, the novelty is limited and the paper is primarily empirical as it does not provide novel theoretical insights.
As such, the presented empirical evaluation is insufficient to convey the benefit of the proposed method, in particular:
(i) apart from MNIST, from real-world datasets results are present solely on CIFAR-10, moreover (ii) these results are incomplete in terms of analysis with varying hyper-parameters (attack radius and the number of attack steps), and (iii)  does not show a clear advantage over existing methods, see above for further comments.

---

> ### Author Response · Authors · 2021-11-23
> **Response to reviewer 7nra (Part 1)**
>
> **While the authors compare with existing methods on CIFAR-10, comparison on other real-world datasets is missing, given that the paper is empirical.**
>
> We agree with the reviewer that comparison on other real-world datasets is an interesting question. However, it is a common choice to consider MNIST and CIFAR-10 for adversarial training in the literature, for example, please see [1,2,3]. In particular, MSD [2] only considers about MNIST and CIFAR-10. Since this paper explores the possibility of building multiple robustness, our major comparison model is MSD. Therefore, we adopted the same datasets as MSD in order to make a fair comparison.
>
> **While the authors perform a thorough analysis on MNIST, to argue in favor of MAT it would be helpful to have such analysis on CIFAR-10 as MNIST can be in some cases misleading. For example, the paper lacks results on CIFAR-10 with: (1) varying radius,**
>
> Thank you for the constructive comment, which allows us to study in particular the robust accuracy against the attack radius of MAT on CIFAR-10 w.r.t that of the state-of-art MSD. We include a more detailed analysis on CIFAR-10 in the revised version (please check Figure. 3 in Section 4.2 of the revised paper). Specifically, we add the following additions to our results:
>
> 1. the performance of MAT is comparable w.r.t MSD on $\ell_{\infty}$ and $\ell_1$- PGD attack with varying attack radius;
>
> 2. MAT outperforms MSD when the attack radius increases (greater than the common choice $0.5$).
>
> **The paper lacks experiments with a varying number of attack steps, in particular, showing its generalization and robustness for different a varying number of steps.**
>
> Good point. Specifically, following the reviewer’s suggestion, we add two experiments with varying number of attack iterations in Section 4.2. Please check Figure 2 for MNIST and Figure 3 for CIFAR-10 in the revised version. Indeed, we can draw the following conclusions:
>
> 1. both variants of our solution, MAT and MAT+MSD, are robust to varying number of attack iterations and the test accuracy remains nearly constant after the attack has converged;
>
> 2. MAT outperforms MSD and MAT+MSD for $\ell_{\infty}$; for  $\ell_2$-PGD attack and $\ell_1$-PGD attack, the performance of MAT+MSD surpasses that of the other two methods on MNIST;
>
> 3. all the solutions (our two variants and MSD) are comparable for CIFAR-10 for varying radius of attack, except for small radius $\ell_2$ and $\ell_{\infty}$ attacks where MSD slightly outperforms.
>
> **Solely two architectures are used, thus it is unclear if MAT performs well using different models.**
>
> Thanks for the comment. We justify our choice of architectures from the following two aspects.
> 1. In Section 6, we noted that our model relies on the large capacity of the model architecture, which allows MAT to adversarially train multiple sub-models simultaneously. Therefore, all we require is a sufficiently large capacity of the architecture.  This explains our choice of architectures for MNIST (resp. CIFAR-10) as ResNet-18 (resp. ResNet-50). We do not use any other features of ResNet-18 (resp. ResNet-50). Thus, they can be replaced by other architectures such as Inception, VGG, etc.
> 2. We choose ResNet-18 (resp. ResNet-50) mainly because it is a common choice, for example, see [1,2,3]. Furthermore, it is unlikely that MAT might fail on different architectures as long as the capacity is large enough because MAT inherits the structure of MIMO and the algorithm of adversarial training. Note that the MIMO structure could be regarded as a special kind of multi-task learning technique, which doesn't require any specific model architectures [6], and adversarial training is widely used on different model architectures [1,4,5]. Therefore, there is no specific requirement on the model architecture for MAT to be effective except for the larger capacity.

---

> ### Author Response · Authors · 2021-11-23
> **Response to reviewer 7nra (Part 2)**
>
> **Focusing on CIFAR-10, the MSD method [20] seems to provide better performances across the different attacks, see Tab.5. The authors acknowledge this and argue that MAT may need further hyperparameter tuning, which raises a concern about how the hyperparameters are selected, as ideally the selection of the presented results should be fair/fixed toward all methods.**
>
> There are two points that need to be clarified in this question. 1. how the hyperparameters are selected; and 2. why MAT may need further hyperparameter tuning.
>
> 1. We adopted the same hyperparameter as the MSD method, including the step size of the PGD attack, attack radius $\epsilon,$ and termination max iteration. That is, all the hyperparameters are chosen as those that provide the best performance for MSD (which may not be the best for MAT) in order to present results on the same baseline.Please check Sec. C1. for detailed information about hyperparameter tuning.
>
> 2. Focusing on CIFAR-10, we agree with the reviewer that the MSD method seems to provide better performances on  $\ell_2$- PGD attack when the test radius is $0.5$.  However, MAT outperforms MSD whenever the $\ell_2$- PGD attack radius is greater than $0.5$. We plot the test accuracy curves of MAT+MSD, MAT and MSD on MNIST under PGD $\ell_\infty$-, $\ell_2$-, and $\ell_1$- attacks with varying attack radius on the CIFAR-10 dataset as the reviewer suggested in Section 4.2 of the revised paper, please see Figure 3 for further information.
>
> **If I am correct, from Tab. 9 in the App. MAT without is less robust to AutoAttack relative to MSD.**
>
> Thanks for the comment. The original Table 9 reported robustness accuracy on CIFAR-10 against AutoAttack (it has been moved from the appendix to the main text as the reviewer suggested in Question 4; please check the revised Table 4 (MNIST) and Table 5 (CIFAR-10) in the revised version. We think the performance between MSD and MAT on CIFAR-10 against AutoAttack is comparable given that there is only a $0.6\%$ ($57.0\%-56.4\%$) difference. Note that MAT outperforms MSD on MNIST against AutoAttack by $25.3\%$ ($82.3\%-57.0\%$). In this case, the improvement over MSD is obvious.
>
> **While the flow is easy to follow, the writing is in some cases informal and thus ambiguous, e.g., in the abstract 'attacks do not transfer easily', and careful proofreading and polishing are needed.**
>
> Thank you for the suggestion. We revised the paper carefully. To be specific, we rewrote Sections 3 and 4 for improving the quality of writing. In particular, to answer the reviewer's comment about defining transferability, we formally explain this in Sec. 4.1.
> In particular for $\ell_\infty$-attacks, we denote
> $N_{ii}\triangleq$ successful attacks to sub-model $i$ out of $1000$ $\ell_{\infty}$-attacks generated from sub-model $i$.
> $N_{ij}\triangleq$ successful attacks to sub-model $j$ out of $1000$ $\ell_{\infty}$-attacks generated from sub-model $i$.
> Then, we define the transferability of adversarial examples from $i$ to $j$ as $r_{ij}\triangleq N_{ij}/N_{ii}.$
> $r_{ij}$ higher indicates the sub-models of MAT (resp. vanilla ensemble) are more robust to $\ell_{\infty}$-attacks, that is, $\ell_{\infty}$-attacks is harder to transfer between the sub-models of vanilla ensemble. We study two kinds of  $\ell_{\infty}$-adversarial examples. The first kind is $\ell_{\infty}$-adversarial attack generated from sub-model 1, from which we can study the transferablility of $\ell_{\infty}$-adversarial attack generated from sub-model 1 by counting the successful attacks to sub-model 2, sub-model 3, and vanilla-ensemble (resp. MAT). Empirically, to explore the diversity among sub-models of the vanilla-ensemble (resp. MAT), the sub-models 1, 2, and 3 have been trained against respectively $\ell_1$, $\ell_2$, and $\ell_\infty$ PGD attacks; then, we evaluate vanilla-ensemble (resp. MAT) against $\ell_{\infty}$-adversarial attacks and report their clean accuracy (the results corresponds to $\ell_{1}$ and $\ell_{2}$ are similar and summarized in appendix G).
>
> **In the introduction the authors argue with certainty that ensemble methods fail due to two main reasons, there should be a citation, or alternatively, elaborate that this is argued in this paper and point to the relevant section. See also the relevant question 4 below.**
>
> The reviewer is right. Thanks for pointing this out. We add [7] in the introduction, to justify the two reasons that ensemble methods fail to defend against adversarial attacks. Furthermore, as suggested by the reviewer, the relevant question 4 can also justify this part. The revised Table 1 shows the vanilla ensemble does not provide extra benefits in terms of robust accuracy compared to each sub-model. We will add this observation in the revised paper. In particular, please check point (2) of the reply to question 4 below for details.

---

> ### Author Response · Authors · 2021-11-23
> **Response to reviwer 7nra (Part 3)**
>
> **It would be helpful to include the total training times per epoch for the different methods for fixed setup.**.
> Thanks for the constructive suggestion.  We add the training times per epoch in the revised version in appendix E. In particular, on MNIST, MSD requires about 13h to train where MAT requires 10h. On CIFAR10, MSD requires 141h23min where MAT requires 123h07min to train. Compared to adversarial training, the extra computation comes from the change of input sizes. The following table is included in appendix E in the revised paper.
>
> **Include detailed captions, e.g. dataset for each table, etc. Similarly for the tables, rather 'main result on...' be more precise, e.g. rows/columns are methods used for training or testing, etc.**
> We thank the reviewer for the constructive comment. We rewrite the captions of every figure and table in the paper. Please see the revised version for details.
>
> **Incomplete sentence in App. C.1**
> Thanks, fixed.
>
> **Sec 4.3: Our methods also increasing $\rightarrow$ increase**
> Thanks, fixed.
>
> **Resnet-50 for CIFAR-10 is less-common, did the authors try with ResNet-18? Moreover, what are the models used by MSD [20]? Ideally, the authors should include a comparison between the two methods using the same model for a fair comparison.**
>
> Good observation.
>
> 1. Reply to the first question: We mentioned in Sec. 6 that our model does require a larger model size. This is due to the MIMO structure, where a large-sized model ensures that MAT could train three sub-models in one neural network simultaneously. We tried MAT using ResNet-18 on CIFAR-10 dataset, and it does not perform well as anticipated due to the limited capacity of ResNet-18.
>
> 2. Reply to the second question: MSD uses ResNet-18 on CIFAR10 and a two-layer neural network on MNIST. However their model performs poorly using ResNet-50 on CIFAR10, and ResNet-18 on MNIST. To be specific, we reproduce their method with their choices of network architectures, and get similar performance as they stated in the paper. However, when we increase the model size to Resnet-50 on CIFAR10, MSD only achieves robust accuracy $14.6\%$ against $\ell_1$ PGD attack while MAT achieves $47.6\%$.  Similarly, their method only achieves $0.9\%$ robust accuracy against $\ell_\infty$ PGD attack using ResNet-18 on MNIST, while our method achieves $89.2\%$ . To conclude, their method performs poorly when using larger models.
>
> Thus, in the paper, we contrast their best results with the smaller models with our best results with larger models. Please check the detailed comparison table through the anonymous [link](https://drive.google.com/drive/folders/1\_1DVlxqT4q-w72a7nGE5Q4qqXFsl6A51).
>
> **For how many epochs are MAT and the baselines trained?**
>
> The MAT, MAT+MSD, as well as the baselines are all trained for 25 epochs for MNIST dataset (please see Sec. C.1 for more detailed information about the epochs and step size choices).  As for the CIFAR10 dataset, the step size choices are summarized in the  Sec. C.1; furthermore, the MAT, MAT+MSD, as well as the baselines are all trained for 75 epochs. Thanks for noting. We will add this information to the revision.
>
> **I would recommend moving the AutoAttack results on CIFAR-10 in the main paper, as such comparison is important. Moreover, how do these AutoAttack results compare to state-of-the-art AT methods (other than MSD)?**
>
> Thanks for the constructive comment. We agree with reviewer that AutoAttack results are important to make a comparison.  We moved the AutoAttack results on MNIST (resp.  CIFAR-10) from the appendix E to Table 4 (resp. Table 5).
>
> Reply to the second comment: The state-of-art for defending*multiple* attacks is MSD. There are state-of-the-art methods for defending *single* adversarial attacks such as [1], however, they only focuses on *one specific* attack. Even though the accuracy for a single attack is higher, previous AT-based methods cannot deal with multiple attacks.

---

> ### Author Response · Authors · 2021-11-23
> **Response to reviewer 7nra (Part 4)**
>
> **Looking solely at Tab.1 could also indicate that the two AT models are robust. What are the clean accuracies and PGD robustness of the models in Tab.1 both separately and the two as an ensemble?**
>
> Thanks for the insightful comment. Indeed, we can capture the clean accuracy and PGD robustness of the models in Table. 1 both separately and the two as an ensemble. We modified the Table. 1 in the revised version as the reviewer suggested. In particular,
>
> 1. In Table. 1, we add the clean accuracy of each model and its ensemble model (the corresponding row is called "no attack"), similarly, in Table. 2, we add the clean accuracy of MAT and its three sub-models;
>
> 2. To explore the diversity among sub-models of vanilla-ensemble (resp. MAT), the sub-models 1, 2, and 3 have been trained against respectively $\ell_1$, $\ell_2$, and $\ell_\infty$ PGD attacks; then, we evaluate vanilla-ensemble (resp. MAT) against $\ell_{\infty}$-adversarial attacks and report their clean accuracy (the results corresponds to $\ell_{1}$ and $\ell_{2}$ are similar and summarized in appendix G). We study two kinds of  $\ell_{\infty}$-adversarial examples. The first kind (corresponds to the first row) is $\ell_{\infty}$-adversarial attack generated from sub-model 1 , from which we can study the transferability of $\ell_{\infty}$-adversarial attack generated from sub-model 1 by counting the successful attacks to sub-model 2, sub-model 3, and vanilla-ensemble (resp. MAT). The second kind (correspondint to the second row) is using $\ell_{\infty}$-adversarial examples generated from vanilla-ensemble (resp. MAT) to attack each sub-model and their vanilla-ensemble (resp. MAT) to evaluate the overall performance of vanilla-ensemble (resp. MAT).  Comparing Table. 1 and Table. 2, we observe the following phenomena: 1) The transferability ratios of MAT are $r_{12}=23.7\%, r_{13}=33.3\%$, which are much lower than  $r_{12}=200\%, r_{13}=153.8\%$ of vanilla ensemble. This indicates the sub-models of MAT are more robust to $\ell_{\infty}$-attacks while $\ell_{\infty}$-attack is easier to transfer between the sub-models of vanilla ensemble. Therefore,  we conclude that the sub-models of MAT are diverse enough to reduce the effective $\ell_\infty$ adversarial transferability.  2) The performance of the vanilla-ensemble is close to the worst sub-model performance. This is indicated by noticing the number of successful $\ell_{\infty}$-attacks for vanilla-ensemble is 960, which is close to $\max(45,994,998)=232$, which means that the vanilla-ensemble does not provide extra robustness compared to its sub-model. However, the successful attacks for MAT $\ell_\infty$-attacks is only 128, which only takes up $23.4\%$ of the worst sub-model performance $\max(114,548,106)=548.$ This suggests that  MAT clearly improves the robustness accuracy compared to that of its sub-models.
>
> **Did the authors compare against PGD-50-10?**
>
> Thanks for pointing out. Indeed, for all the evaluations, we used 10 restarts and we ran 200 iterations instead of 50 iterations to guarantee the convergence of the attacks. Please check the appendix C.2 for detailed hyperparameter choices.  For further clarification, we add Figure 2 (resp. Figure. 3) for MNIST (resp. CIFAR-10) on accuracy against different numbers of iterations of the attacks in Sec. 4.2 in the revised version.
>
> [1] Madry, Aleksander, et al. "Towards deep learning models resistant to adversarial attacks." arXiv preprint arXiv:1706.06083 (2017).
>
> [2] Maini, Pratyush, Eric Wong, and Zico Kolter. "Adversarial robustness against the union of multiple perturbation models." International Conference on Machine Learning. PMLR, 2020.
>
> [3] Tramer, Florian, and Dan Boneh. "Adversarial training and robustness for multiple perturbations." arXiv preprint arXiv:1904.13000 (2019).
>
> [4] Rebuffi, Sylvestre-Alvise, et al. "Fixing data augmentation to improve adversarial robustness." arXiv preprint arXiv:2103.01946 (2021).
>
> [5] Zhang, Hongyang, et al. "Theoretically principled trade-off between robustness and accuracy." International Conference on Machine Learning. PMLR, 2019.
>
> [6] Havasi, Marton, et al. "Training independent subnetworks for robust prediction." arXiv preprint arXiv:2010.06610 (2020).
>
> [7] Yang, Zhuolin, et al. "TRS: Transferability Reduced Ensemble via Encouraging Gradient Diversity and Model Smoothness." arXiv preprint arXiv:2104.00671 (2021).

---

> > ### Comment · Reviewer_7nra · 2021-12-09
> > **Thanks for the thorough response**
> >
> > I would like to thank the authors for their thorough responses.
> >
> > While the authors have improved their submission, some of my main concerns remain, for example:
> > 1. Given that the paper is primarily empirical (and that MNIST is considered by recent AT papers somewhat toy setup), adding results on a different dataset other than CIFAR-10, different model, etc are necessary. I personally would recommend improving the theoretical understanding that motivates the proposed method, e.g. to me it is surprising that baseline methods do not work well with larger models (see below), or what is the theoretical motivation of ensemble methods in terms of improved adversarial robustness and/or generalization. However, if this is difficult, I think that adding more extensive empirical evaluation that is clearly presented (benchmarking methods on setups with equal computation budget, flops, and training time) is necessary.
> > 2. If the baselines use a notably smaller model, I think that the comparison is somewhat unfair, or alternatively, the results should be carefully presented. The authors justify this by claiming the baselines do not work well with larger models on the same dataset, but it is unclear if this is due to lack of hyperparameter tuning, the flow of the baseline method itself, etc. Moreover, if I understood well, the proposed method does not work well for the smaller scale setup on which the baseline works well. This indicates that the authors should be very clear what is the scenario for which they recommend their method (e.g. if computation and model size is not an issue), and justify that the gains are significant--proportionally to the added computational overhead.
> >
> > In summary, I think this write-up (in the current state) while offering an interesting observation, at the same time it raises several questions which seem (to me) either not answered or not clearly presented, and thus I am not confident recommending acceptance.

---

### Decision · Program_Chairs · 2022-01-20

**Decision:**

Reject

**Comment:**

This paper proposes a new ensemble training method for improving adversarial robustness to multiple attacks (e.g., $\ell_2$, $\ell_1$ and $\ell_\infty$). Specifically, authors adopt the recent Multi-Input Multi-Output (MIMO) ensemble architecture for computational efficiency. Then, the authors construct the adversarial examples using the outputs of multiple attacks simultaneously. With these examples, standard adversarial training is conducted on MIMO ensemble.

All reviewers are on the negative side. AC agrees with reviewers’ concerns on limited novelty and insufficient empirical evaluation. AC also thinks that the improvement is not that significant compared to the existing method, especially concerning the real-world dataset. Overall, AC recommends rejection.